# AlphaFold-SFA: Accelerated sampling of *cryptic pocket* opening, *protein-ligand* binding and *allostery* by AlphaFold, slow feature analysis and metadynamics

**Shray Vats[1], Raitis Bobrovs[2], Pär Söderhjelm[3], Soumendranath Bhakat[4]** *

1 Department of Computer Science, University of Texas at Austin, Austin, TX, United States of America, 2 Latvian Institute of Organic Synthesis, Riga, Latvia, 3 Division of Biophysical Chemistry, Chemical Center, Lund University, Lund, Sweden, 4 AlloTec, St. Louis, MO, United States of America

* sbhakat@allotec.bio, bhakatsoumendranath@gmail.com

## Abstract

Sampling *rare events* in proteins is crucial for comprehending complex phenomena like cryptic pocket opening, where transient structural changes expose new binding sites. Understanding these rare events also sheds light on protein-ligand binding and allosteric communications, where distant site interactions influence protein function. Traditional unbiased molecular dynamics simulations often fail to sample such rare events, as the free energy barrier between metastable states is large relative to the thermal energy. This renders these events inaccessible on the timescales typically simulated by unbiased molecular dynamics, limiting our understanding of these critical processes. In this paper, we proposed a novel unsupervised learning approach termed as *slow feature analysis* (SFA) which aims to extract slowly varying features from high-dimensional temporal data. SFA trained on small unbiased molecular dynamics simulations launched from AlphaFold generated conformational ensembles manages to capture rare events governing cryptic pocket opening, protein-ligand binding, and allosteric communications in a kinase. Metadynamics simulations using SFA as collective variables manage to sample 'deep' cryptic pocket opening within a few hundreds of nanoseconds which was beyond the reach of microsecond long unbiased molecular dynamics simulations. SFA augmented metadynamics also managed to capture conformational plasticity of protein upon ligand binding/unbinding and provided novel insights into allosteric communication in receptor-interacting protein kinase 2 (RIPK2) which dictates protein-protein interaction. Taken together, our results show how SFA acts as a dimensionality reduction tool which bridges the gap between AlphaFold, molecular dynamics simulation and metadynamics in context of capturing rare events in biomolecules, extending the scope of structure-based drug discovery in the era of AlphaFold.

**Data Availability Statement:** All necessary input files are accessible here: https://osf.io/wm6vx/. Jupyter notebooks containing necessary analysis can be accessed here: https://github.com/sbhakat/

AlphaFold-SFA/tree/main. Software to perform slow feature analysis on molecular dynamics trajectory can be found here: https://github.com/svats73/md-sfa-msm/tree/main.

**Funding:** The author(s) received no specific funding for this work.

**Competing interests:** Authors declare no conflict of interests. Soumendranath Bhakat and Shray Vats are cofounders of AlloTec Bio, Inc.

## Introduction

The challenge of accurately predicting the 3D structure of proteins based solely on their amino acid sequences has been a longstanding puzzle in the realm of structural biology. Traditionally, scientists relied heavily on experimental techniques like X-ray crystallography and cryogenic electron microscopy (Cryo-EM) to decipher these protein structures [1]. While these methods remain crucial for examining intricate biomolecules, the landscape underwent a transformative shift in 2021. This change was marked by the introduction of AlphaFold [2], an artificial intelligence (AI)-driven model that showcased its prowess in predicting protein 3D structures from their sequences. Building on this innovation, ColabFold [3] was subsequently developed, optimizing AlphaFold to operate seamlessly on Google Colab, thereby democratizing AI-based protein structure prediction. However, it's essential to highlight a shared limitation across AlphaFold, X-ray crystallography, and Cryo-EM: while they excel at capturing a static representation or 'snapshot' of a protein, they fail to capture the conformational dynamics of biologically significant protein movements. Such movements, like the unveiling of cryptic pockets [4] or allosteric communication [5], often involve the sampling of transient high-energy states, commonly referred to as *rare events* [6, 7]. Cryptic pockets have emerged as a frontier in modern drug discovery, offering new opportunities to target proteins previously considered "*undruggable.*" These hidden binding sites are not apparent in the protein's apo state but become accessible upon interaction with small molecules or during *rare* conformational transitions. Sampling cryptic pocket opening in apo proteins is of significant interest in drug discovery, as targeting these pockets has the potential to yield highly selective and potent inhibitors with unique structural features distinct from catalytic sites or known binding interfaces. The strategy of targeting cryptic pockets has proven attractive in developing selective inhibitors against challenging targets, such as plasmepsin-II, a key enzyme in antimalarial drug discovery. Similar to cryptic pockets, allostery in protein kinases is a rare conformational event which represents a complex signaling pathway via which changes in the active site of kinase induces conformational changes at a distal site. Understanding allostery in kinases is key to uncovering the mechanisms by which these enzymes interact with protein partners and regulate downstream signaling across several disease pathways. Our study aims to capture conformational allostery in Receptor-Interacting Protein Kinase 2 (RIPK2), a serine-threonine kinase implicated in immune and inflammatory responses. It functions as a key signaling molecule in the nucleotide-binding oligomerization domain (NOD)-like receptor pathways, which are vital for recognizing intracellular pathogens and danger signals [8, 9]. Upon activation, RIPK2 undergoes ubiquitination, a process significantly facilitated by the E3 ubiquitin ligase, XIAP (X-linked Inhibitor of Apoptosis Protein). This interaction between RIPK2 and XIAP is pivotal in modulating downstream signaling pathways, especially the nuclear factor-kappa B (NF-κB) and mitogen-activated protein kinase (MAPK) pathways, which are essential for the production of pro-inflammatory cytokines and immune responses. Elucidating allosteric fingerprints in RIPK2 is key to gain insights into kinase function beyond catalytic activity, potentially revealing new strategies for therapeutic intervention in various pathological conditions.

Molecular dynamics (MD) simulations [10, 11] in theory, possess the capability to sample *rare events*. Yet, a significant limitation emerges; the timescales that MD simulations can access are restricted. As a result, the molecular system frequently remains trapped in a specific free energy minimum. This confinement means it's challenging to capture the infrequent rare events with the desired level of detail. Enter a new trick: the stochastic subsampling of multiple sequence alignment (MSA). This technique recently enabled AlphaFold to sample a diverse conformational ensemble of 3D protein structures [12, 13], even those in high energy states.

Seeding MD simulations with these AlphaFold generated ensembles followed by Markov State Modelling (MSM) can provide Boltzmann-weighted probability distribution associated with cryptic pocket opening [14]. However, this approach isn't without its challenges. To gather sufficient data in order to construct a reliable MSM [15–17], one would require a combined simulation time spanning tens of microseconds. Enhanced sampling techniques, especially metadynamics, offer a potent alternative to sample rare transitions within reasonable timescale. Metadynamics accelerates sampling by depositing Gaussian shaped bias along predefined reaction coordinates often known as collective variables [18, 19]. Selecting the optimal collective variables [7], capable of capturing rare transitions, remains an active area of research.

In this paper, we introduce slow feature analysis (SFA) [20], an unsupervised learning algorithm which can capture slowly varying features from high-dimensional temporal data generated by MD simulations. SFA trained on short MD simulations seeded from AlphaFold generated ensemble is used as collective variables in metadynamics simulations to capture cryptic pocket opening and protein-ligand binding in plasmepsin-II [21, 22], a drug target for malaria. We have also shown how SFA can capture *allosteric hotspots* associated with receptor-interacting protein kinase 2 (RIPK2) mediated protein-protein interaction [9], a key checkpoint associated with inflammatory responses (**Fig 1**). We propose slow feature analysis as a dimensionality reduction algorithm that bridges the gap between AlphaFold and

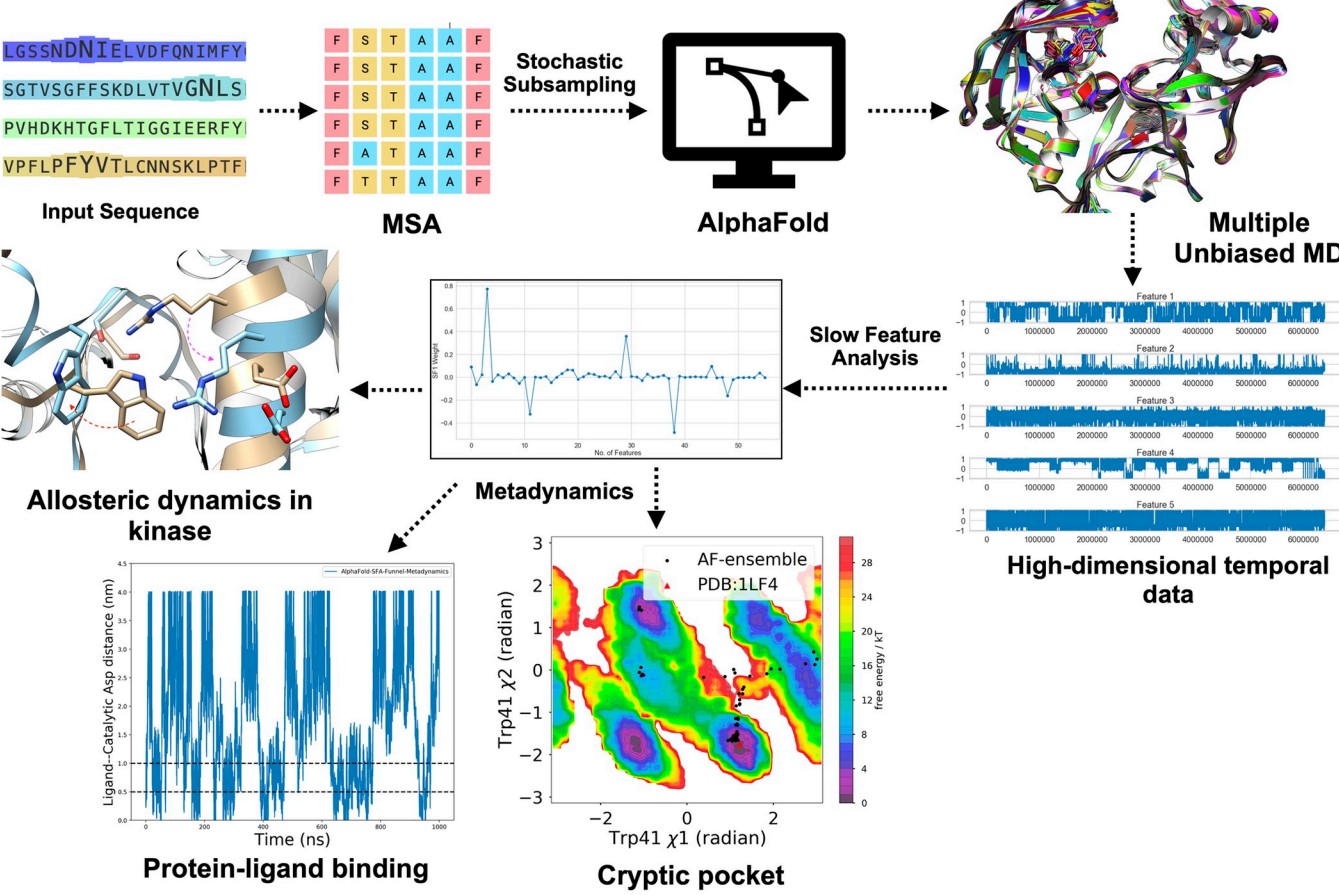

**Fig 1. Combining AlphaFold, MD simulations, slow feature analysis (SFA) and metadynamics to sample *rare* bimolecular dynamics.**

metadynamics and manages to capture Boltzmann distribution associated with a wide range of biological phenomena such as cryptic pocket opening, protein-ligand binding and identifying allosteric hotspots associated with kinase mediated protein-protein interactions.

Multiple sequence alignment (MSA) of an input sequence followed by stochastic subsampling enabled AlphaFold to generate a conformational ensemble of plasmepsin II and RIPK2 with structural diversity. AlphaFold generated structural ensemble was used as a starting point to run multiple short unbiased MD simulations which generated high-dimensional temporal data associated with protein dynamics. Slow feature analysis (SFA) captures *slowly varying* features from high-dimensional temporally evolving data. SFA as the collective variables in metadynamics simulations, efficiently sample *cryptic* pocket opening and protein-ligand binding in plasmepsin-II as well as allosteric conformational dynamics in RIPK2.

## Computational methods

**Structural Ensemble generation using AlphaFold.** Structural ensemble for a given sequence was generated using ColabFold implementation of AlphaFold(link:https://colab.research.google.com/github/sokrypton/ColabFold/blob/main/AlphaFold2.ipynb#scrollTo=KK7X9T44pWb7) as described previously by *Meller and coworkers* [14]. We generated initial multiple sequence alignment (MSA) using the MMseqs2 method [23] implemented with ColabFold. We then stochastically subsampled the MSA to a maximum of 32 cluster centers and 64 extra sequences (noted as *max_msa* = 32:64). For the generation of the structural ensemble, we opted for the "*complete*" pairing strategy, which only pairs sequences with a full taxonomic match. We set the number of random seeds to 16 and enabled model dropout. Using dropout, combined with the increased number of random seeds, allows AlphaFold's neural network to tap into the model's inherent uncertainties leading to generation of structural ensemble (total 80 structures for plasmepsin-II) with conformational heterogeneity.

Conformational ensemble of RIPK2 was generated using an older version of ColabFold (https://colab.research.google.com/github/sokrypton/ColabFold/blob/main/beta/AlphaFold2_advanced.ipynb). A structural ensemble consisting of 32 structures of RIPK2 were generated using the following settings as described previously: *msa_method*: *MMseqs2*, *pair_mode* = *unpaired+paired*, *pair_cov* = 25, *pair_qid* = 20, *max_msa* = 32:64, *subsample_msa* = *True*, *num_models* = 1, *num_samples* = 32, *num_ensemble* = 1, *max_recycles* = 3, *is_training* = *True*.

The specific sequences used to generate the conformational ensemble are detailed in the 'Choice of sequences for structure prediction' in S1 Text.

**Molecular dynamics simulations.** Each structure from the conformational ensemble generated by AlphaFold was prepared for molecular dynamics simulations using the *tleap* module from Amber2022 [24], following the protocol outlined by *Meller et al* [14]. In brief, the proteins were parameterized with the AMBER FF14SB force field [25]. To achieve system neutrality, 17 Na$^+$ ions were added to each system. Systems were then solvated within a truncated octahedron box with TIP3P [26] waters, ensuring a minimum of 10 Å between the protein and the edge of the box. The system underwent a two-phase minimization: (a) an initial phase where only the water and ions were minimized while the protein was held in place using a restraint potential of 100 kcal/mol$^{-1}$ Å$^2$ (200 steepest descent steps followed by 200 conjugate gradient steps), and (b) an unrestrained minimization of the entire system over 500 steps.

After the minimization process in Amber2022, we transformed Amber topologies into Gromacs format via Acpype [27]. Each system was gradually heated from 0 to 300 K for 500 ps in an NVT ensemble, with harmonic restraints (500 kJ mol$^{-1}$nm$^{-2}$) on the backbone's heavy atoms. Subsequently, systems were equilibrated for 200 *ps* in an NPT ensemble at 300 K,

devoid of restraints. The *Parrinello–Rahman* barostat [28] ensured a consistent pressure of 1 bar, while the v-rescale thermostat controlled the temperature. Production runs were performed in the NPT ensemble, maintaining conditions at 300 K and 1 bar. The leapfrog integrator and Parrinello–Rahman thermostat was employed with a 2 fs timestep. Nonbonded interactions had a cutoff of 1.0 nm and long-range electrostatic interactions were treated using the Particle Mesh Ewald (PME) method [29] with a 0.16 nm grid spacing. The LINCS algorithm [30] was used to constrain covalent bonds with hydrogen atoms. Heating, equilibration, and production runs were performed using Gromacs 2022 [31].

For the 80 structures of plasmepsin-II produced by AlphaFold, we conducted two independent 40ns production runs, each with unique initial starting velocities. For 32 structures of RIPK2, we performed ten independent 20 ns productions runs each with different initial velocities. Trajectories were saved every ps.

Unbiased molecular dynamics simulation of RIPK2+XIAP complex (PDB: 8AZA [32]) was also performed using a similar protocol. The missing residues of apo RIPK2 (PDB: 5AR2 [33]) and RIPK2+XIAP complex were modelled using Modeller web server [34] integrated with UCSF Chimera.

**Slow feature analysis (SFA).** Slow Feature Analysis [20] (SFA) is a dimensionality reduction technique designed to process high-dimensional, temporally evolving data. Its primary goal is to transform a J-dimensional input signal, $c(t)$, using a set of nonlinear functions, $g_k(c)$, to produce output signals $y_k(t) = g_k(c(t))$. These output signals are optimized to minimize $\Delta y_k = \langle \dot{y}_k^2 \rangle$, where $\dot{y}$ represents the derivative of $y$, and $\langle \rangle$ indicates temporal averaging. This minimization targets the extraction of features that vary slowly over time. SFA also imposes additional constraints as follows:

a. each output feature must have zero mean $\langle y_k \rangle = 0$,

b. unit variance $\langle y_k^2 \rangle = 1$, and

c. decorrelated from others $\langle y_k y_{k'} \rangle = 0$ for all $k' < k$

These constraints ensure that each extracted feature is scaled similarly, uncorrelated with others, and avoids the trivial solution, $y_k = c$ where $c$ is a constant. When detailing the algorithm below, signals represented by capitals represent raw signals, while signals represented by lower-case represent normalized signals.

To perform the SFA algorithm, first start with a J-dimensional input signal $X(t)$. Next, normalize the input signal to get:

$$c(t) = [c_1(t) \ldots c_j(t)]^T \tag{1}$$

where:

$$c_j(t) = \frac{C_j(t) - \langle C_j \rangle}{\sqrt{\langle (C_j(t) - \langle C_j \rangle)^2 \rangle}} \tag{2}$$

Such that:

$$\langle c_j \rangle = 0 \text{ and } \langle c_j^2 \rangle = 1$$

Next, perform a linear or a non-linear expansion using a set of functions $H(c)$ to produce an expanded signal $Z(t)$. In the case of a quadratic expansion this would include monomials of degree one and degree two including mixed terms as shown below, which would result in

quadratic SFA:

$$H(c) = [c_1, \ldots, c_j, c_1 c_1, c_1 c_2, \ldots, c_j c_j] \tag{3}$$

and thus:

$$Z(t) = H(c(t)) = [c_1(t), \ldots, c_j(t), c_1(t)c_1(t), c_1(t)c_2(t), \ldots, c_j(t)c_j(t)] \tag{4}$$

The next phase involves sphering (or whitening) of $Z(t)$, generating a normalized signal $z(t)$ through the equation:

$$z(t) = S(Z(t) - \langle Z \rangle) \tag{5}$$

where:

$$\langle z \rangle = 0 \text{ and } \langle zz^T \rangle = I$$

In this case, $I$ is the identity covariance matrix and $S$ is a sphering matrix. Matrix $S$ can be solved by performing PCA on matrix $(Z(t) - \langle Z \rangle)$.

Further, PCA is applied to the matrix $\langle \dot{z} \dot{z}^T \rangle$ where $\dot{z}$ is the time derivative of the sphered expanded signal $z(t)$. The $K$ eigenvectors with the lowest eigenvalues $\lambda_k$ result in the normalized weight vectors $w_k$ satisfying:

$$w_k : \langle \dot{z} \dot{z}^T \rangle w_k = \lambda_k w_k \text{ where } \lambda_1 \leq \lambda_2 \ldots \leq \lambda_k \tag{6}$$

This leads to the formulation of the desired set of real-valued functions $g(c) = [g_1(c), \ldots, g_k(c)]^T$ where:

$$g_k(c) = w_k^T \cdot h(c) \tag{7}$$

and:

$$h(c) = S(H(c) - \langle Z \rangle) \tag{8}$$

Here, $S$ is the sphering matrix that was earlier solved to normalize the expanded signal $Z(t)$. Now, define the output signal $y(t)$ as:

$$y(t) = g(c(t)) \tag{9}$$

where:

$$\langle y \rangle = 0, \langle yy^T \rangle = I, \text{ and } \Delta y_k = \langle \dot{y}_k^2 \rangle = \lambda_k$$

In this formulation, the components of y(t), representing the extracted slow features, are characterized by zero mean, unit variance, and mutual decorrelation, thereby encapsulating the core principles of SFA.

**Application of SFA on molecular dynamics.** Slow feature analysis was performed on training data generated from unbiased molecular dynamics simulations launched from the conformational ensemble of plasmepsin II (80 structures * 2 clones each * 40 ns = 6.4 microsecond) and RIPK2 (32 structures * 10 clones each * 20 ns = 6.4 microsecond) generated by AlphaFold.

In this study, we adopted linear Slow Feature Analysis (SFA) for our experimental analysis, implemented in the *sklearn SFA* package (https://sklearn-sfa.readthedocs.io/en/latest/). Our initial step involved the compilation of featurised trajectory data into a J-dimensional input signal, denoted as $c(t)$ and normalizing the data by solving for the whitening matrix $S$ for the input signal using PCA and subsequently transforming it into $c^{white}$. Whitening can be

expressed as the linear map $c^{white} = D^{-\frac{1}{2}}U^T c$ where the covariance matrix of $c(t)$ is decomposed as $C_c = UDU^T$.

We then do PCA on finite differences of the whitened input signal $\dot{c}_t^{white} = c_{t+1}^{white} - c_t^{white}$ to get the decomposition of the covariance matrix for the finite differences $\dot{c}$ such that $C_{\dot{c}} = V\Lambda V^T$.

The eigenvectors corresponding to the smallest eigenvalues in this decomposition represented the normalized weights of the slowest moving features. These extracted weights, derived from the training data, were then utilized to construct a new collective variable for subsequent metadynamics simulations. This approach leverages the strength of linear SFA in distilling critical dynamical features from high-dimensional data, enabling nuanced exploration and modeling of the underlying processes.

**Markov state model.** Markov state model was performed on sin and cos transformed $\chi1$ and $\chi2$ angles of Trp41 extracted from unbiased molecular dynamics simulations (80 structures * 2 independent clones * 100ns each = total 16 μs) launched from AlphaFold generated conformational ensemble of plasmepsin-II. *Kmeans* clustering was performed on the transformed dihedral space with *k = 200*. Finally, maximum likelihood MSM was generated using a lag time of 6 ns (**S2 Fig** highlights the implied timescale plot). Equilibrium populations extracted from MSM is projected along different features to highlight conformational heterogeneity associated with plasmepsin II. PCCA+ [35] was used to generate macrostate definition which manages to distinguish conformational states associated with Trp41 (**S2 Fig**). MSM generation was performed using PyEMMA 2.5.7 [36].

**Metadynamics.** Well-tempered metadynamics simulations were performed using the first two slow features as collective variables (CVs) at 300K. Gaussian width was chosen by taking *1/3* of the standard deviation of first two slow features respectively using first 10000 frames of the training data. For plasmepsin-II, gaussians of height 1.50 kJ/mol were deposited at every 500 steps. The Gaussian widths for the first two slow features were set at 0.32 and 0.25, determined by taking approximately one-third of the standard deviation from the unbiased training data. The bias-factor was set at 20. We performed two distinct metadynamics simulations, each lasting around ~400 ns. One began from the closed state (PDB: 1LF4) [37] and the other from the cryptic pocket open state (PDB: 2BJU) [38]. For RIPK2 (PDB: 5AR2), 500 ns well-tempered metadynamics simulation was performed using first two slow features as CVs using a gaussian widths of 0.16 and 0.25, height 1.50 kJ/mol and bias factor 20. Metadynamics simulations were performed using Gromacs 2022 patched with Plumed 2.7 [39]. Unbiased free energy surfaces along different features were extracted using the reweighting protocol developed by *Tiwary and Parrinello* [40].

Funnel metadynamics [41, 42] represents an enhanced sampling technique designed to improve sampling efficiency of protein-ligand binding. This method specifically targets the sampling process along the pathway of ligand unbinding/binding. It employs a unique, funnel-shaped restraint potential, effectively narrowing down the exploration space when the ligand is in its unbound state leading to rapid binding/unbinding of ligand. Funnel metadynamics simulations with and without SFA was performed using a small molecule bound to 'cryptic pocket' of plasmepsin II (PDB: 7QYH). The funnel potential is highlighted in **Fig 2**. The following parameters were used to define the funnel: $Z_{cc}$ = 2.0 nm, $R_{cyl}$ = 0.1 nm, and $\alpha$ = 0.5 radian.

Traditional funnel metadynamics was performed using the distance between the center of mass (COM) of ligand and C$\gamma$ atoms of catalytic Asp dyads (Asp 34 and 214) as CV. A steep repulsive wall was applied along the COM distance with a spring constant of 50000 kJ/mol/nm$^2$. Other details regarding the funnel setup have been described in the Supporting

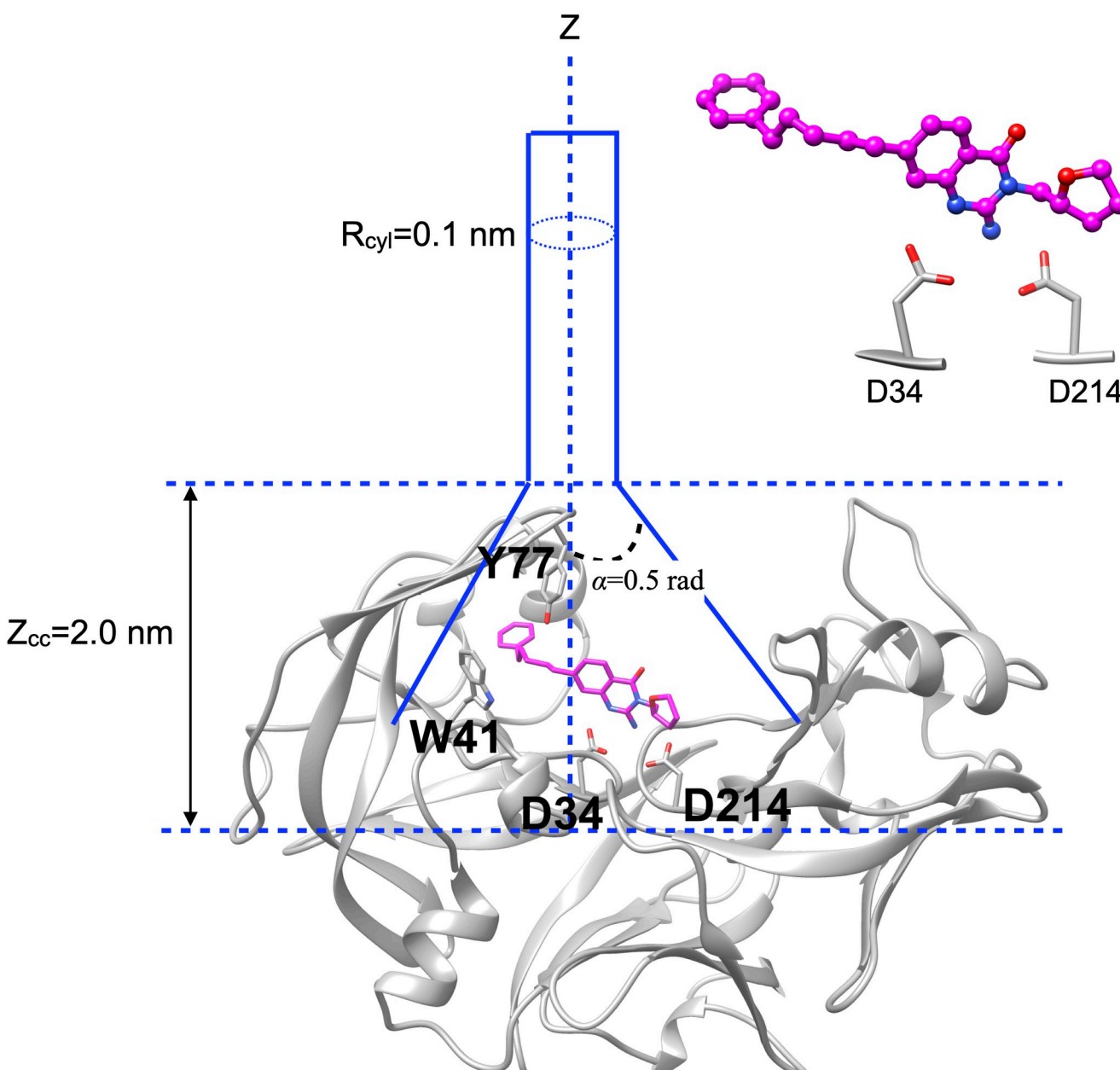

**Fig 2. Depiction of the funnel setup employed in metadynamics simulations.** Central to this setup is the measurement of the distance between the center of mass of the catalytic aspartate dyads (specifically D34 and D214) and the ligand. This serves as the ligand binding/unbinding collective variable (CV) in traditional funnel metadynamics. A small molecule (highlighted in *magenta*) bound to the cryptic pocket of plasmepsin II (PDB: 7QYH) was used for funnel metadynamics simulations.

information. Finally, well-tempered metadynamics was performed using a Gaussian height of 1.5 kJ/mol, width of 0.03 nm and bias factor of 20. SFA-Funnel metadynamics was performed using first two slow features as additional CVs as described in the previously. Funnel metadynamics in principle can predict protein-ligand binding free energy. In our study, given the lack of experimental binding free energy data, we aimed to determine whether SFA-augmented funnel metadynamics can accelerate ligand binding.

## Results

### SFA captures critical fluctuations necessary for cryptic pocket opening

SFA trained on small independent unbiased molecular dynamics simulation launched from a structural ensemble generated by AlphaFold managed to capture flipping of Trp41 in plasmepsin II, necessary for cryptic pocket opening (**S3 and S4 Figs**). It also captured flipping of Tyr77 along $\chi 1$ angle as another key feature (**S3 Fig**). Flipping of Tyr77 in conjunction with flipping of Trp41 exposes a fully 'open' cryptic pocket primed for ligand binding (**Fig 3**).

A recent study highlighted how microsecond long unbiased MD simulation launched from apo plasmepsin II failed to capture cryptic pocket opening [14]. To test the effectiveness of SFA as CVs with metadynamics we performed two independent simulations, one launched from 'closed' state (PDB: 1LF4) and the other from the cryptic pocket 'open' state (PDB: 2BJU, removing the ligand to make it apo). Metadynamics simulations using first two slow features as CVs accelerated the sampling of the conformational space (**S5 Fig**) and managed to sample multiple flipping events along Trp41 $\chi 1$ and $\chi 2$ angles leading to close $\leftrightarrow$ open transitions in plasmepsin II within ~350–400 ns of simulation time. It is key to highlight unbiased MD simulations launched from either *apo-like* 'open' and 'closed' state failed to sample open $\leftrightarrow$ closed transitions (**Fig 4**). Further, reweighted free energy surfaces along Trp41 $\chi 1$ and $\chi 2$ angles agreed extremely well with the sampling from MSM developed using a total 16 μs simulation data generated by Meller and coworkers (**S1 Fig**). This highlight accelerated sampling of metadynamics using SFA as CVs in capturing rare event such as *cryptic pocket opening* in plasmepsin II. Comparison of reweighted free energy surfaces at different time intervals from metadynamics simulations highlighted the convergence (**S6 Fig**) for the choice of force field and water model.

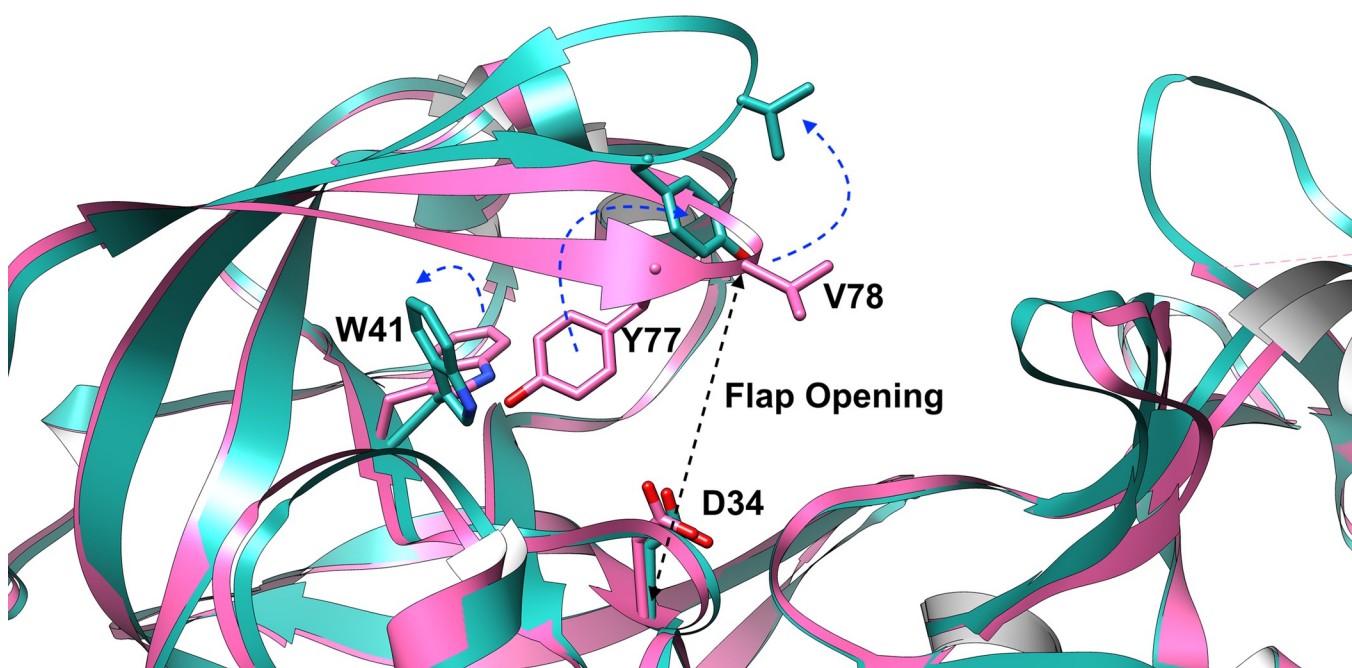

**Fig 3. Key structural features of plasmepsin II.** Structural features of plasmepsin II highlighting differences between closed (magenta, PDB: 1LF4) and open states (blue, PDB: 2BJU). 'Flap opening' is the defined by the $C\alpha$—$C\alpha$ distance between Asp34 and Val78. Flipping of Tyr77 and Trp41 (highlighted using blue arrow) opens a deep cryptic pocket in plasmepsin II.

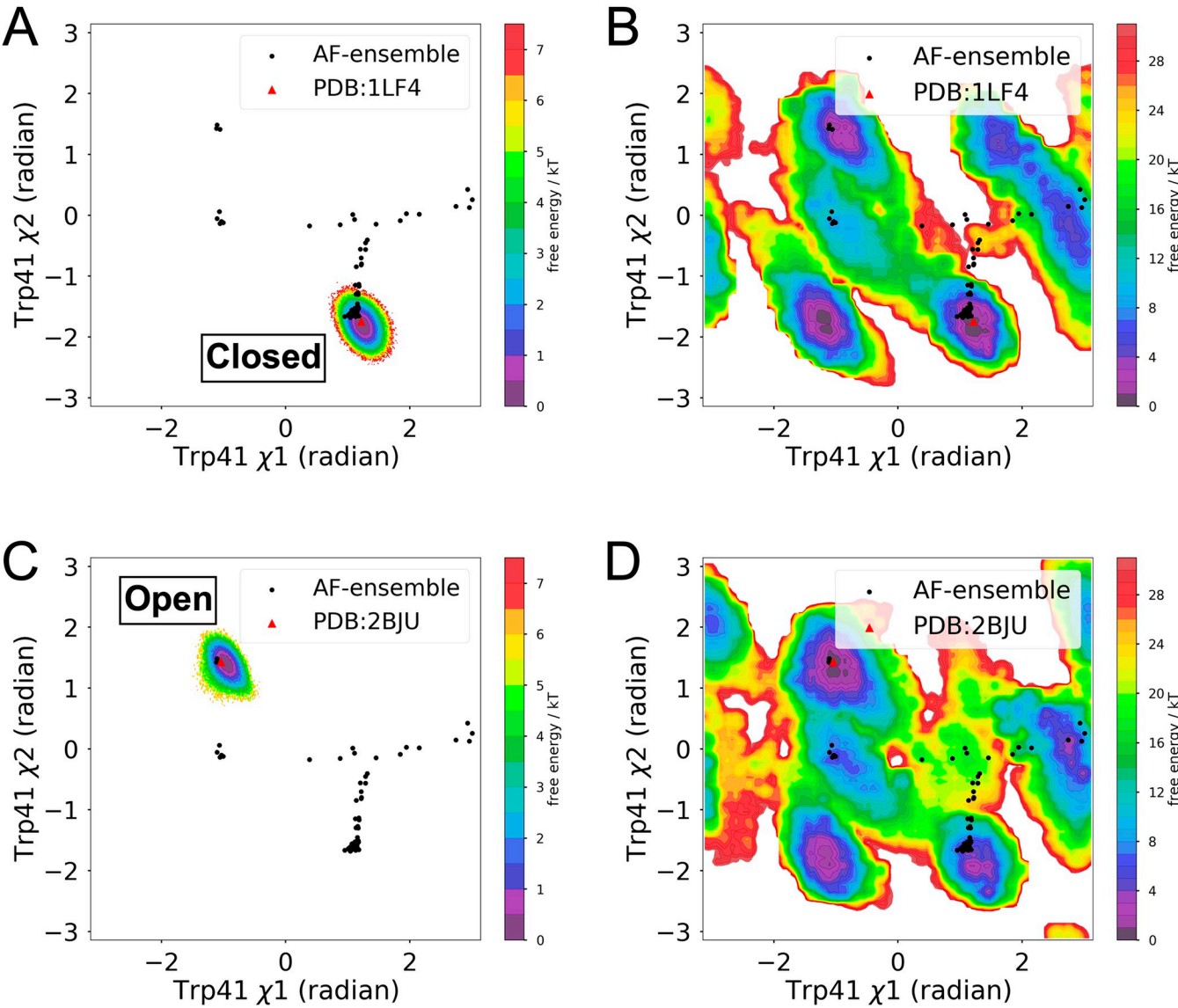

**Fig 4. Metadynamics simulations using first two slow features as CVs captures cryptic pocket opening in plasmepsin II.** (A) Free energy surface projected along Trp41 $\chi1$ and $\chi2$ angles from unbiased MD simulations started from closed state. (B) Reweighted free energy surface from metadynamics simulation started from closed state projected along Trp41 $\chi1$ and $\chi2$ angles. (C) Free energy surface projected along Trp41 $\chi1$ and $\chi2$ angles from unbiased MD simulations started from open state. (D) Reweighted free energy surface from metadynamics simulation started from open state projected along Trp41 $\chi1$ and $\chi2$ angles. It is important to note that the simulation length for the metadynamics simulations is 500 ns each whereas the unbiased molecular dynamics simulations started from closed (PDB 1LF4) and open (PDB: 2BJU) states has an aggregate simulation time of ~16 µs each (160 replicas * 100 ns each). AlphaFold generated structural ensembles are highlighted in *black dots*.

Metadynamics also captured flipping of Tyr77 (**S8 Fig**) which is a key residue governing the flap 'opening' in plasmepsin II as described by *Bhakat and Söderhjelm* [22]. Opening of the flap in conjunction with flipping of Trp41 exposes the 'deep' cryptic pocket (**Fig 5 and S7 Fig**).

Metadynamics using SFA CVs also managed to capture an alternate flipped state of Trp41 which has not been sampled by AlphaFold. Such a state has been captured by holo crystal structure of plasmepsin II, PDB: 4Z22 [43] (**S9 Fig**).

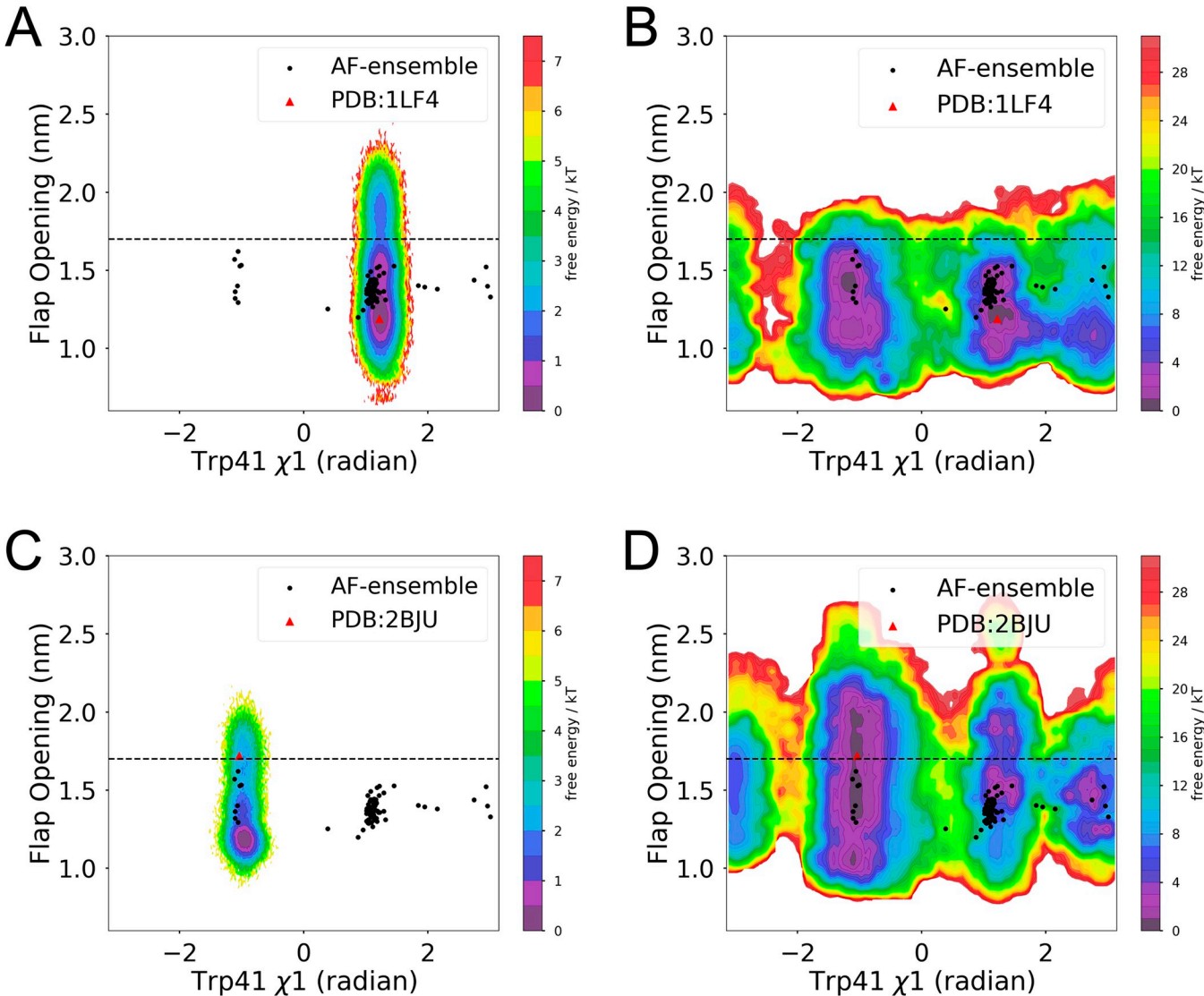

**Fig 5. SFA-metadynamics managed to capture a deep cryptic pocket opening in plasmepsin II.** (A) Free energy surface projected along flap opening and Trp41 $\chi1$ angle from unbiased MD simulations starting from closed state. (B) Reweighted free energy surface from metadynamics with SFA CVs starting from closed state projected along flap opening and Trp41 $\chi1$ angle. (C) Free energy surface projected along flap opening and Trp41 $\chi1$ angle from unbiased MD simulations starting from open state. (D) Reweighted free energy surface from metadynamics with SFA CVs starting from open state projected along flap opening and Trp41 $\chi1$ angle. Flap opening is defined by the $C\alpha$—$C\alpha$ distance between Asp34 and Val78 as highlighted in Fig 3. Flap opening in conjunction with Trp41 flipping exposes a deep cryptic pocket in plasmepsin II. The black dotted line demarcates flap opening distance of 1.7 nm. Flap opening distance > 1.7 nm in conjunction with flipping Trp41 ($\chi1$ = -1 radian) corresponds to deep cryptic pocket opening.

## SFA augmented funnel metadynamics accelerates ligand binding in plasmepsin II

Ligand binding in the cryptic pocket of the plasmepsin-II requires correct orientation of Trp41 as well as opening of the flap which allows ligand entry. Previous study by Bobrovs and coworkers highlighted the importance of incorporating flipping of Trp41 and flap opening as CVs in metadynamics to accurately sample multiple binding/unbinding events of a small molecule. The aim of funnel metadynamics investigations in our case was to highlight the ability of SFA in capturing protein dynamics of plasmepsin II which is critical for sampling ligand

binding to the cryptic pocket of plasmepsin II. We performed two sets of funnel metady-namics: a) traditional funnel metadynamics with distance between center of mass of the ligand and catalytic aspartic acid dyads as a CV and b) SFA augmented funnel metadynamics which uses first two slow features trained on AlphaFold seeded MD simulations as orthogonal CVs in conjunction with distance CV (see Methods).

SFA augmented funnel metadynamics managed to capture multiple ligand binding events compared to traditional funnel metadynamics. Moreover, SFA-funnel metadynamics captures opening of the flap as well as flipping of Trp41, two key conformational events require for ligand binding (**Fig 6**). *Bobrovs and co-workers* [44] highlighted the necessity of using dynam-ical information associated with flap opening via 'path' as orthogonal CVs to capture acceler-ated ligand binding when compared with traditional funnel metadynamics. Funnel metadynamics simulation using SFA as CVs sampled multiple recrossing associated with flap opening, revealing a *deep cryptic* pocket, necessary for ligand binding (**Fig 7**). The flap opening is critical for ligand binding in plasmepsin-II. SFA-funnel metadynamics not only accelerates ligand binding but also provides a novel alternative to path CVs in capturing protein dynamics in plasmepsin II.

## SFA captures allosteric hotspots in RIPK2

The RIPK2-XIAP interaction represents a critical checkpoint in the regulation of inflamma-tion and innate immunity, highlighting its potential as a therapeutic target in diseases charac-terized by excessive or chronic inflammation [8, 9]. The interaction between RIPK2 and XIAP is intricately regulated by the activation loop and allosteric modulation in RIPK2 [32]. The activation loop, a part of RIPK2's kinase domain, undergoes conformational changes upon activation, influencing its interaction with XIAP via allostery (**Fig 8**). However, due to its dynamical nature, the activation loop in RIPK2 often remain unresolved in X-ray crystallogra-phy, posing challenges in comprehending how the activation loop's conformational diversity allosterically influences its interaction with XIAP.

SFA trained on AlphaFold seeded MD simulation managed to capture several critical conformational fluctuational associated with RIPK2 (**S10 Fig**). One of the key structural fea-tures captured by SFA is the flipping of Phe165 in the DFG moiety. Metadynamics simula-tion using first two slow features as CVs managed to capture Phe165 flipping in apo RIPK2, a rare event which has not been sampled in ~500 ns unbiased MD simulation (**Fig 9**). It is important to highlight that such flipping has only been observed (**S14 Fig**) in holo RIPK2 bound with small molecules, GSK583 (PDB: 5J7B [45]), SB-203580 (PDB: 5AR4 [33]), pyra-zolocarboxamide scaffold (PDB: 6SZJ [46]) and CSLP18 (PDB: 6FU5). SFA also managed to capture flipping of Trp170, a key residue in present in the activation loop acting a hallmark of allostery mediated conformational transition in RIPK2. SFA-metadynamics allowed us to sample a high-resolution picture of the allosteric hotspot involving Trp170. In the active state of RIPK2, Trp170 points towards αC helix which is represented by distance between Glu68-CD—Trp170-NE1. Another structural feature of the active RIPK2 is the Arg65 side-chain mediated H-bond interaction involving Ser168 (captured by the distance between Arg65 and Ser168) (**S15 Fig**). This H-bond interaction keeps the stabilized the activation loop which facilitates interaction with XIAP (**S17 Fig**). SFA-metadynamics managed to cap-ture rare transition between active <—> inactive states of RIPK2. A hallmark of such tran-sition is the flipping of Trp170 which breaks the packing of Trp170 and αC helix. This change is quantified by the distance between Glu68-CD and Trp170-NE1 (**Figs 10, Fig 11**). This further lead to breaking of H-bond interaction involving Arg65 and Ser168 (**S15 Fig**) which shifts the αC helix in an 'outward' conformation (**Fig 11 and S16 Fig**). Transition of

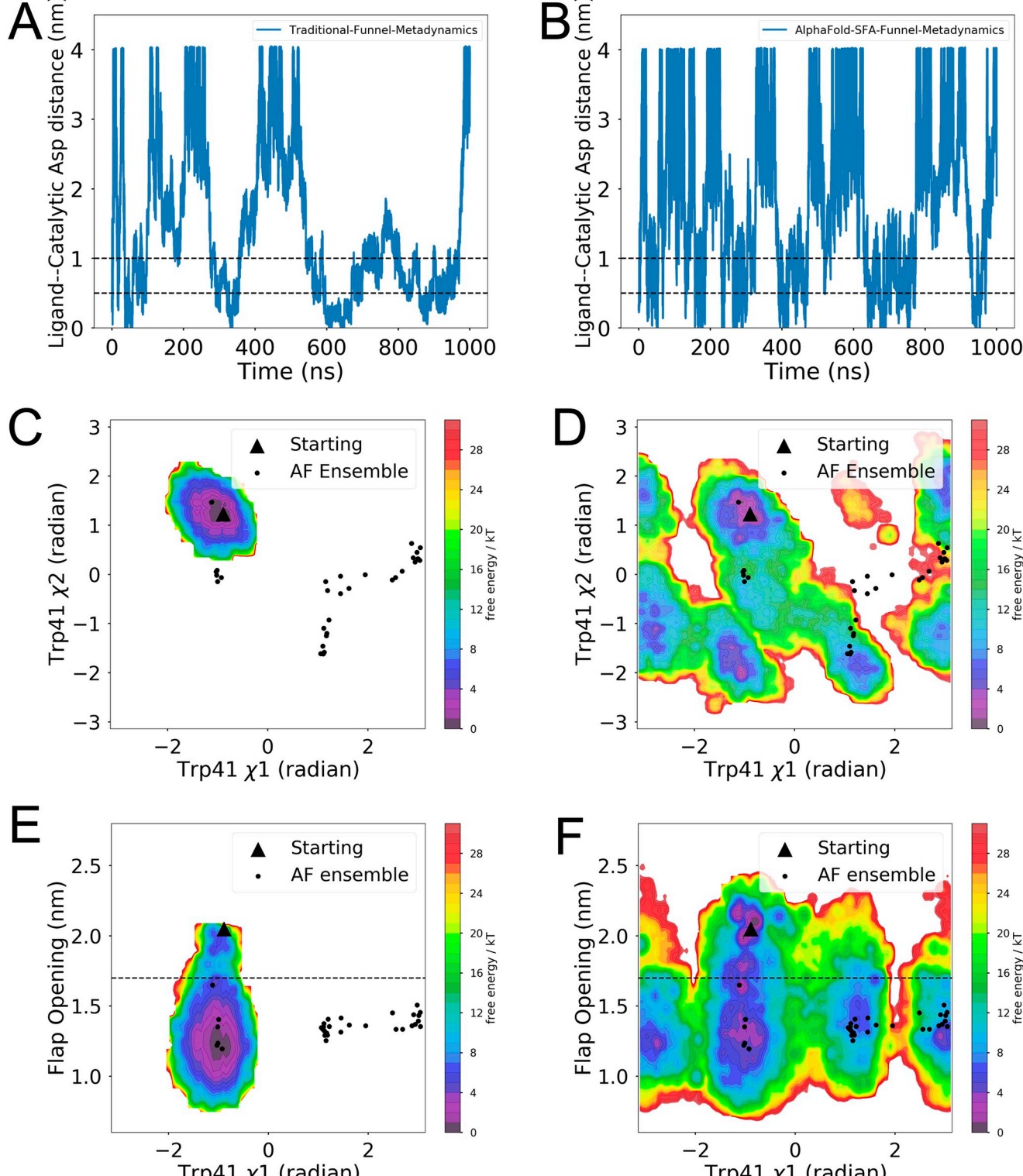

**Fig 6. Metadynamics simulation using slow features as CVs capture protein dynamics necessary for multiple ligand binding/unbinding.** (A) Time trace of the distance between ligand and the COM of the catalytic aspartic acids from traditional funnel metadynamics simulation. (B) Time trace of the distance between ligand and the COM of the catalytic aspartic acids from SFA augmented funnel metadynamics simulation. (C) Reweighted free energy surface projected along Trp41 $\chi1$ and $\chi2$ angles from traditional funnel metadynamics. (D) Reweighted free energy surface projected along Trp41 $\chi1$ and $\chi2$ angles from SFA augmented funnel metadynamics. (E) Reweighted free energy surface projected along flap opening and Trp41 $\chi1$ angle from traditional

funnel metadynamics. (F) Reweighted free energy surface projected along flap opening and Trp41 $\chi$1 angle from SFA augmented funnel metadynamics. Ligand binding in the deep cryptic pocket is indicated by a ligand-catalytic Asp distance of less than 0.5 nm. A distance greater than 0.5 nm but less than 1.0 nm suggests the ligand is in an intermediate region, maintaining contact with the protein. A distance exceeding 1.0 nm in (A) and (B) implies the ligand is completely unbound from the protein. Additionally, incorporating slow features as orthogonal CVs in funnel metadynamics successfully captured two rare conformational transitions: the flipping of Trp41 (D) and the opening of the flap (F) when compared to traditional funnel metadynamics (C, E), accelerating ligand binding/unbinding.

αC helix to an 'outward' conformation shifts the Glu66 and breaks the salt-bridge interaction involving Lys47 and Glu66 (**Fig 12**).

Such conformational transitions destabilize the activation loop which blocks XIAP binding. Molecular simulation of RIPK2-XIAP complex highlighted a critical H-bond interaction involving Arg171 of RIPK2 and Asp214 of the XIAP (**S17E and S18** Figs). This interaction is dependent on the stabilization of the activation loop involving Trp170 and Arg65 (**S17C and S17D** Figs). Overall, our study highlighted how SFA trained on AlphaFold seeded molecular simulation augmented by metadynamics captures previously elusive allosteric hotspot involving activation loop of RIPK2 which governs RIPK2+XIAP interactions.

## Discussion

Stochastic subsampling of MSA allowed AlphaFold to sample a structural ensemble of plasmepsin II with conformational diversity; however, it failed to sample the 'deep' cryptic pocket opening (**Fig 4**). Well-tempered metadynamics simulations with first two slow features as CVs managed to capture 'deep' cryptic pocket opening in plasmepsin II within a factor of total simulation length when compared to MSM based approach. Deep cryptic pocket opening is a rare event in plasmepsin II which is a combination of flipping of Trp41 and flap opening [47]. SFA-augmented funnel metadynamics outperformed traditional funnel metadynamics in sampling the complex dynamics of ligand binding and unbinding to the cryptic pocket of plasmepsin II. This approach successfully captured multiple recrossing associated with ligand binding and enabled more effective conformational sampling of the protein dynamics. Specifically, it facilitated the sampling of crucial conformational transitions like the deep cryptic pocket

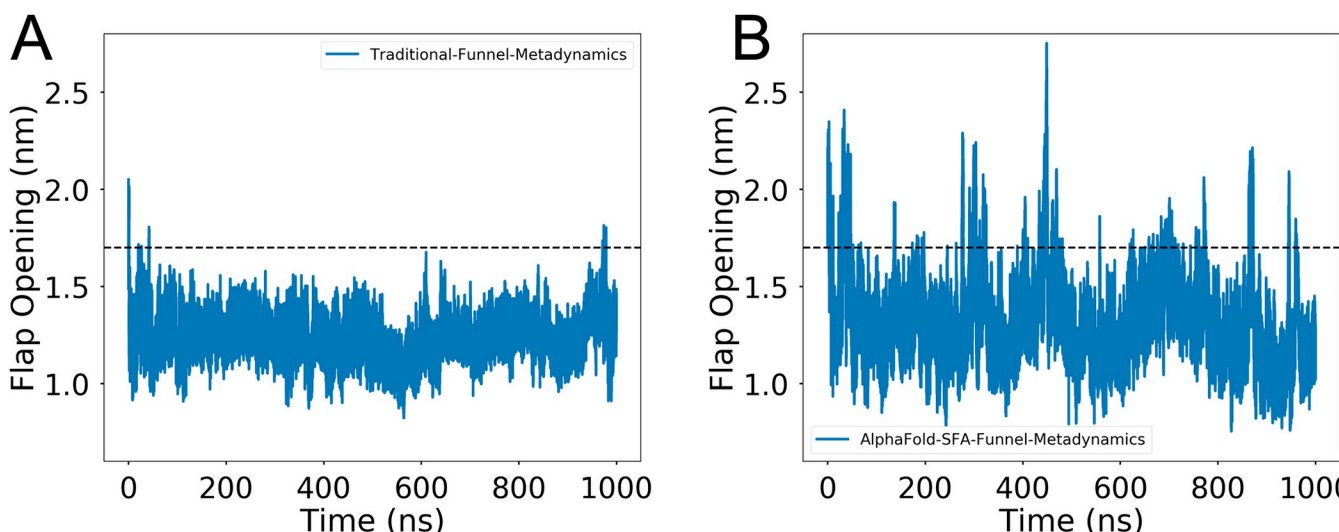

**Fig 7. SFA augmented funnel metadynamics capture flap opening in plasmepsin II.** (A) Time trace of flap opening in traditional funnel metadynamics. (B) Time trace of flap opening in SFA augmented funnel metadynamics. SFA augmented funnel metadynamics sample deep cryptic pocket opening necessary for ligand rebinding. The black dotted line demarcates flap opening distance of 1.7 nm. Flap opening distance > 1.7 nm signifies deep cryptic pocket opening.

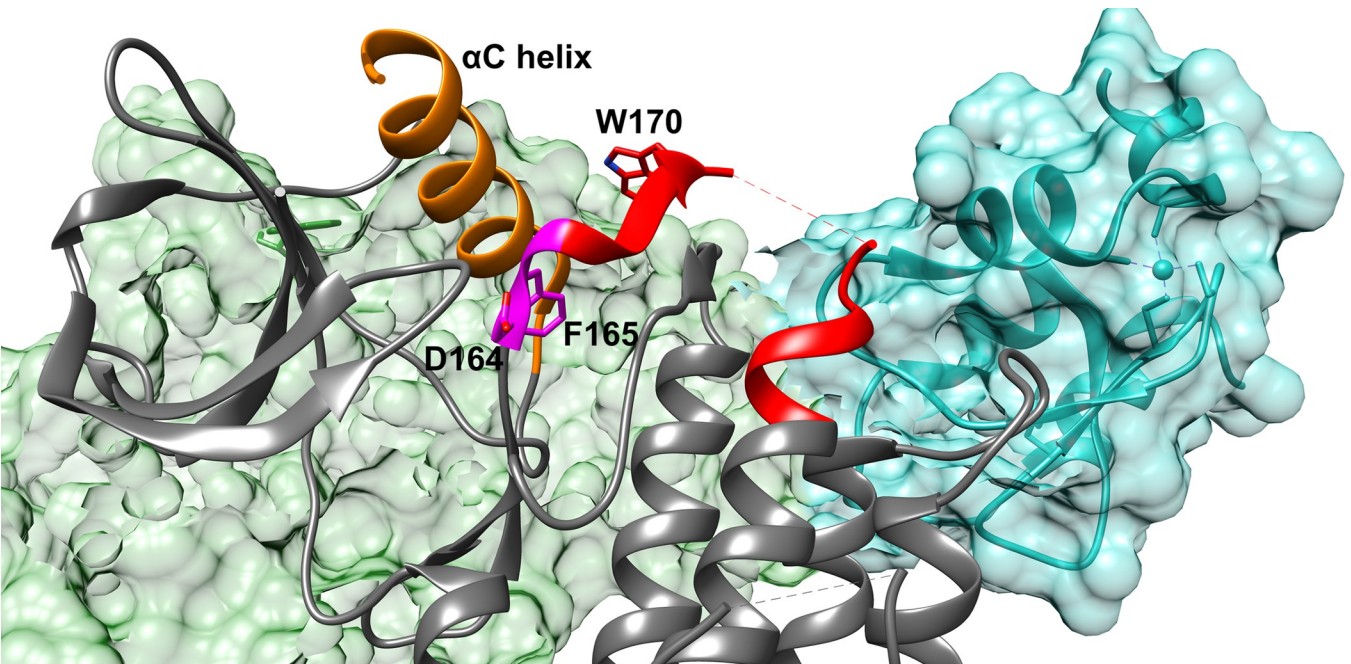

**Fig 8. Cryo-EM structure (PDB: 8AZA) of RIPK2 and XIAP.** In this structure, XIAP is visualized with a cyan surface and is located away from the RIPK2's active site. The active site of RIPK2, characterized by the DFG motif (highlighted in magenta), engages with XIAP through the kinase's activation loop, depicted in red. Notably, sections of the activation loop remain unresolved in both cryo-EM and X-ray structures of RIPK2, indicating dynamic nature of the region. The αC helix (highlighted in orange), is posited to significantly influence RIPK2's conformational changes by its interactions with the activation loop. However, the precise mechanisms by which the αC helix and activation loop affect RIPK2's conformation and its interaction with XIAP are not yet fully understood and remain an *open question*. It is important to note that the PDB structures of apo and holo RIPK2 can be divided into two categories: those in which the orientation of Trp170 is resolved, and those in which it is not. The second monomer of RIPK2 is highlighted in green surface representation.

opening and the flipping of Trp41, both vital for ligand binding. In contrast, traditional funnel metadynamics without SFA did not adequately sample these transitions, notably failing to capture the deep cryptic pocket opening and Trp41 flipping, thus impeding effective ligand binding. Additionally, in the absence of SFA, traditional funnel metadynamics sampled formation of hydrogen bond formation between Tyr77 and Asp34 (**S10 Fig**), as noted by *Bhakat and Söderhjelm* [22]. Formation of such hydrogen bond interaction blocks the flap of plasmepsin-II from opening which is critical for ligand binding. Remarkably, SFA trained on AlphaFold-seeded MD simulations, was able to detect the flipping of Tyr77, a critical movement for enabling flap opening [22]. Utilizing the first two slow features as collective variables in funnel metadynamics significantly enhanced the sampling efficiency of flap opening, thereby promoting ligand binding.

SFA trained on AlphaFold-seeded MD simulations, effectively identifies allosteric hotspots in RIPK2. The dynamics associated with these hotspots are crucial for the interaction between RIPK2 and XIAP, a key element in NOD1/2-mediated immune responses, with implications in inflammatory bowel disease and other conditions. Our study reveals how the conformational dynamics of the αC helix, activation loop and the Asp-Phe-Gly (DFG) loop in RIPK2 regulate active <—> inactive transition (**Fig 13**). Active to inactive transition disrupts the interaction between RIPK2 and XIAP, crucial for NOD-mediated inflammatory signaling. It is important to highlight that conformational ensemble generated by AlphaFold failed to sample dihedral flipping associated with Phe165 in RIPK2 but did sample dynamics of activation loop and αC helix as evidenced by orientation of Trp170 and Arg65 (**S19 Fig**).

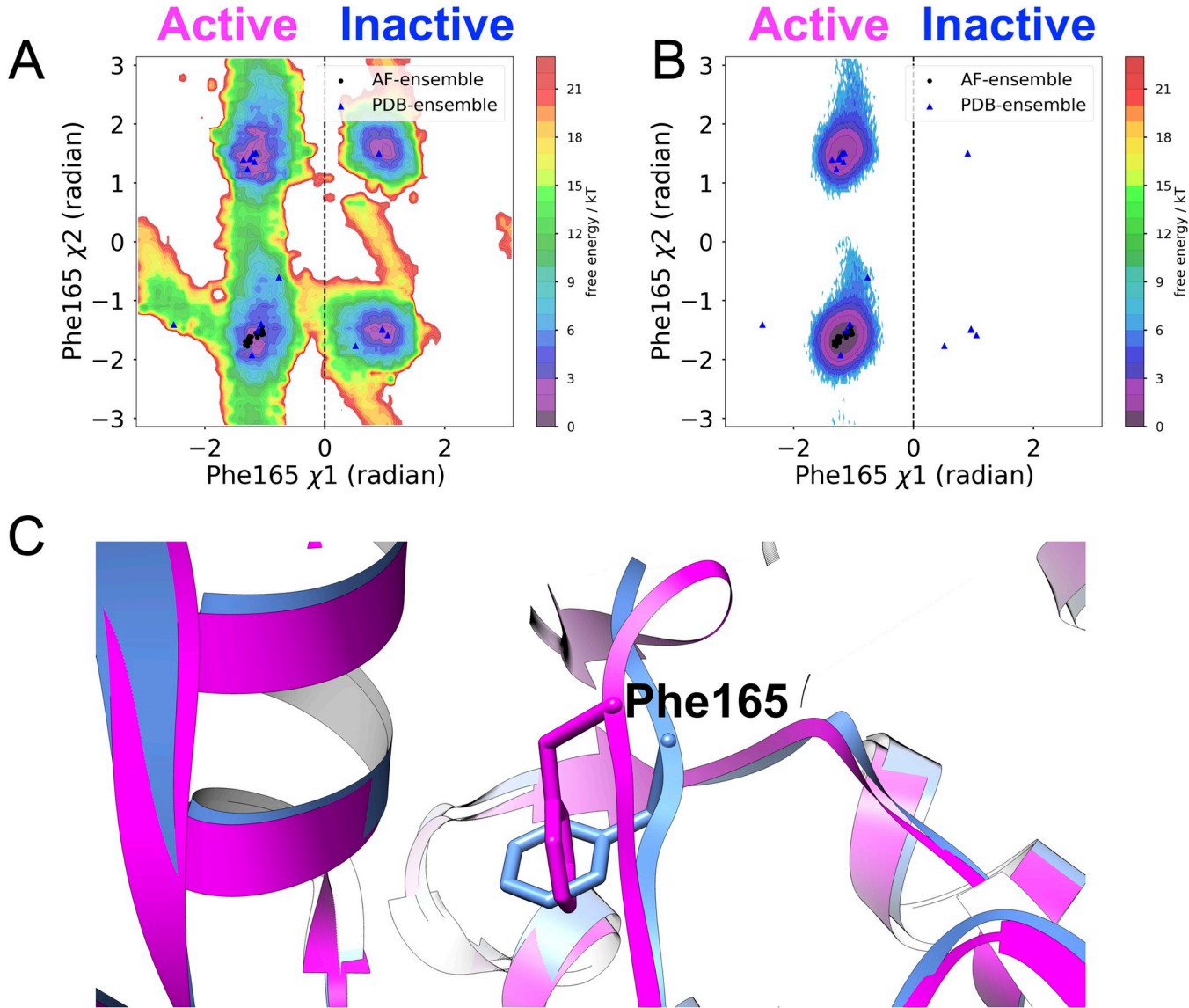

**Fig 9. SFA metadynamics captures active and inactive conformation in RIPK2.** (A) Reweighted free energy landscape corresponding to the $\chi 1$ and $\chi 2$ angles of Phe165, as sampled in 500 ns of SFA-metadynamics. (B) Free energy surface projected along Phe165 $\chi 1$ and $\chi 2$ angles from unibiased MD simulation of same length. (C) Structural orientation of Phe165 in active (magenta) and inactive (blue) states of RIPK2 shows the flipping along $\chi 1$ angle. SFA-metadynamics successfully sampled the transition from active to inactive states of Phe165, a process not captured in the unbiased MD simulations of both apo RIPK2 and RIPK2-XIAP (**S17**B Fig). The black dots indicate values of Phe165 dihedral angles in AlphaFold generated conformational ensemble. The black dotted line demarcates Phe165 $\chi 1$ angle = 0 radian.

SFA-augmented metadynamics successfully captures two vital interactions: 1) hydrogen bond interactions between the Arg65 sidechain and Ser168, monitored by the distance between Arg65 and Ser168, and 2) the interaction involving Trp170 and residues in the αC helix, where Trp170's side-chain NH group is oriented towards Glu68, measured by the distance between Trp170 and Glu68. This finding expands upon *Pellegrini et al.*'s report of a water-mediated hydrogen bond between Trp170 and Glu68 [48]. These interactions are pivotal in defining RIPK2's *active state,* stabilizing the activation loop, which serves as a binding platform (*'landing pad'*) for XIAP. Notably, the H-bond interaction of Arg171 in the activation loop with Asp214 in XIAP is essential for stabilizing the RIPK2-XIAP complex.

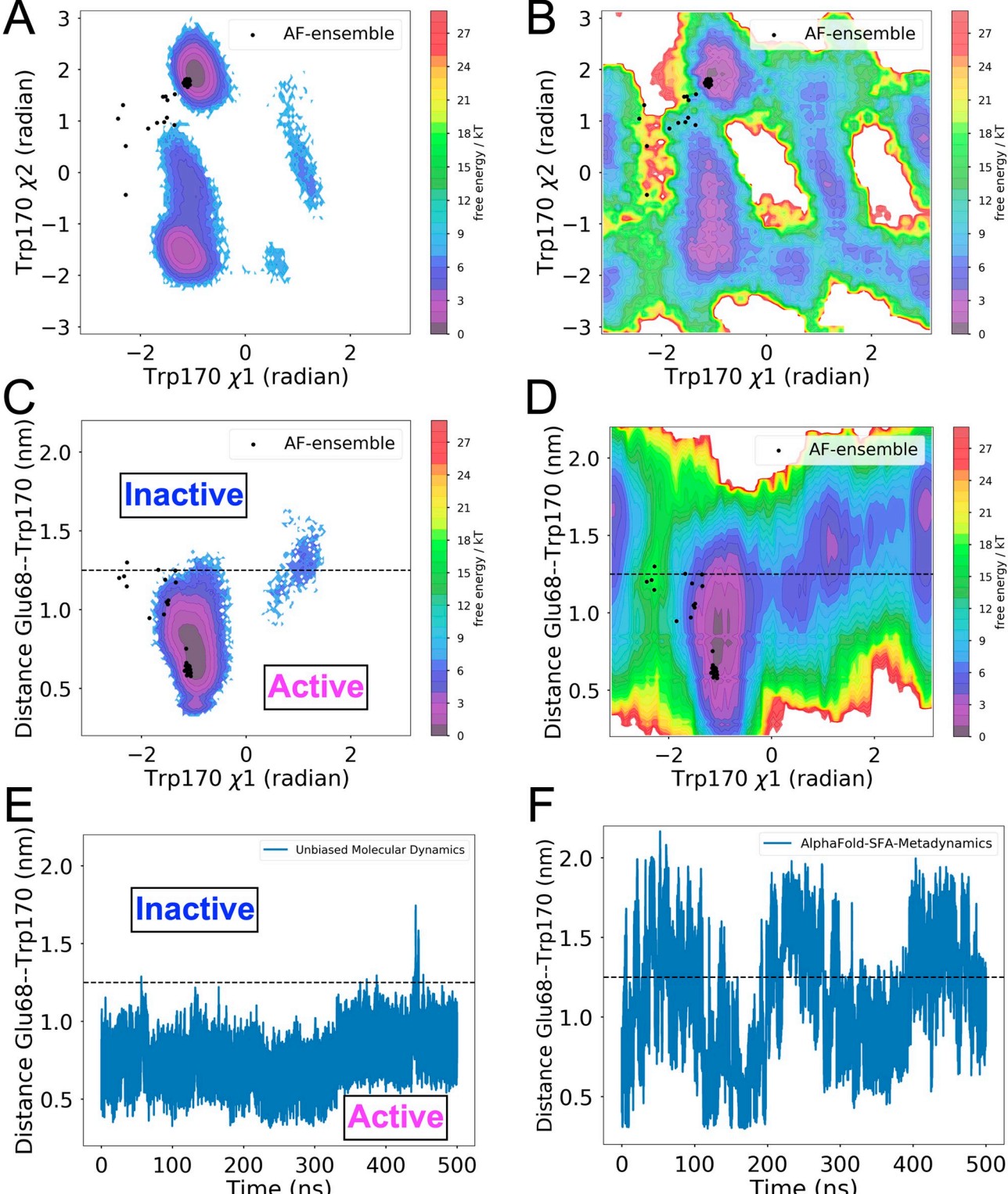

**Fig 10. Flipping of Trp170 around the $\chi 1$ angle is a key structural element for the transition from *active* to *inactive* states in apo RIPK2.** (A) Free energy surface projected along Trp170 $\chi 1$ and $\chi 2$ angles as sampled in unbiased MD. (B) Reweighted free energy surface projected along Trp170 $\chi 1$ and $\chi 2$ angles from SFA metadynamics simulation. (C) Free energy surface projected along the distance between Trp170 and Glu68 and Trp170 $\chi 1$ angle from unbiased MD simulation. (D) Reweighted free energy surface projected along the distance between Trp170 and Glu68 and Trp170 $\chi 1$ angle from SFA metadynamics simulation. (E) Time trace of the Glu68—Trp170 distance sampled in unbiased MD simulation. (F) Time trace of Glu68—Trp170

distance sampled in SFA metadynamics. Flipping of Trp170 disrupts the structural integrity between the activation loop and the αC helix, a change tracked by the distance between Trp170 and Glu68. SFA metadynamics effectively captures the variety of transitions between active and inactive states of apo RIPK2 compared to unbiased MD simulation. Unbiased MD simulation of RIPK2-XIAP complex (S17 Fig) samples active state of RIPK2. AlphaFold generated conformational ensembles are highlighted in black dots. The black dotted line demarcates Glu68—Trp170 distance of 1.25 nm which separates the active and inactive states.

The transition to the *inactive state* involves the disruption of these interactions, increased conformational entropy of the activation loop, and the flipping of Trp170. This flipping breaks the interactions between Trp170 and the αC helix, as well as between Arg65 and Ser168, thus destabilizing the activation loop. Concurrently, the flipping of Phe165 in the DFG loop shifts the αC helix outward and increases the mobility of the activation loop, dismantling the structural integrity of the '*landing pad*,' impeding XIAP binding. Outward conformation of αC helix breaks the salt bridge interaction involving Lys47 and Glu66. Careful analysis of another serine/threonine kinase, BRAF also highlights the coupled motion involving DFG-Phe *flipping* and '*outward*' conformation of αC helix which leads to breaking of salt bridge interaction involving lysine and glutamic acid (S20 Fig). Flipping of Phe of the DFG motif, outward conformation of αC helix and broken salt-bridge interaction between Lys47 and Glu66 are the three key features of Src-like inactive conformation [49] in kinases. SFA-metadynamics successfully captured conformational transitions between active and Src-like inactive conformations, highlighting the effectiveness of our computational protocol in capturing functionally relevant states in RIPK2. Additionally, SFA highlights the significance of the flipping motions of Ile208 and Lys209 in RIPK2's regulatory region. Lys209 forms crucial interactions with Glu211 (S21 Fig) in XIAP. The flipping of Ile208 and Lys209 disrupts these interactions, contributing to the inhibition of XIAP binding. This aligns with experimental findings that indicate the mutation of Lys209 impedes the RIPK2-XIAP interaction [32].

A careful examination of 3D structures of small molecules bound to RIPK2, as deposited in the Protein Data Bank (PDB), reveals two distinct categories: (a) structures in which the position of Trp170 remains unresolved, and (b) structures where Trp170 is oriented towards Glu68 (hallmark of active RIPK2). Notably, in structures where Trp170's position is indeterminate, Phe165 appears in a 'inactive' state, as evidenced in PDB entries 5J7B, 5AR4, 6SZJ and 6FU5 (Fig 9A). This led us to propose that the binding of small molecules to RIPK2 induces conformational changes, transitioning it from an active to inactive state, resulting in RIPK2--XIAP inhibition.

SFA trained on molecular dynamics simulations seeded from AlphaFold structures captures Boltzmann-weighted structural features necessary to sample conformational transitions associated with cryptic pocket opening in plasmepsin II and allosteric dynamics in RIPK2. In the case of cryptic pocket opening, initiating multiple short unbiased molecular dynamics simulations from the diverse conformational ensemble generated by AlphaFold confers an advantage in conformational sampling, as this ensemble captures multiple degrees of freedom along key structural features (Figs 4 and 5).

Conversely, for RIPK2, AlphaFold alone failed to sample conformational transitions across key structural features involved in active-to-inactive transitions (Figs 9, 10, and Fig 12). SFA trained solely on the AlphaFold-generated ensemble would have been unable to capture the critical transitions required for active-to-inactive transitions. However, short molecular dynamics simulations initiated from the AlphaFold-generated ensemble of RIPK2 which sampled structural heterogeneity of the activation loop (missing in the crystal structures of apo and holo RIPK2), captures Boltzmann-weighted structural features governing active-to-inactive transitions.

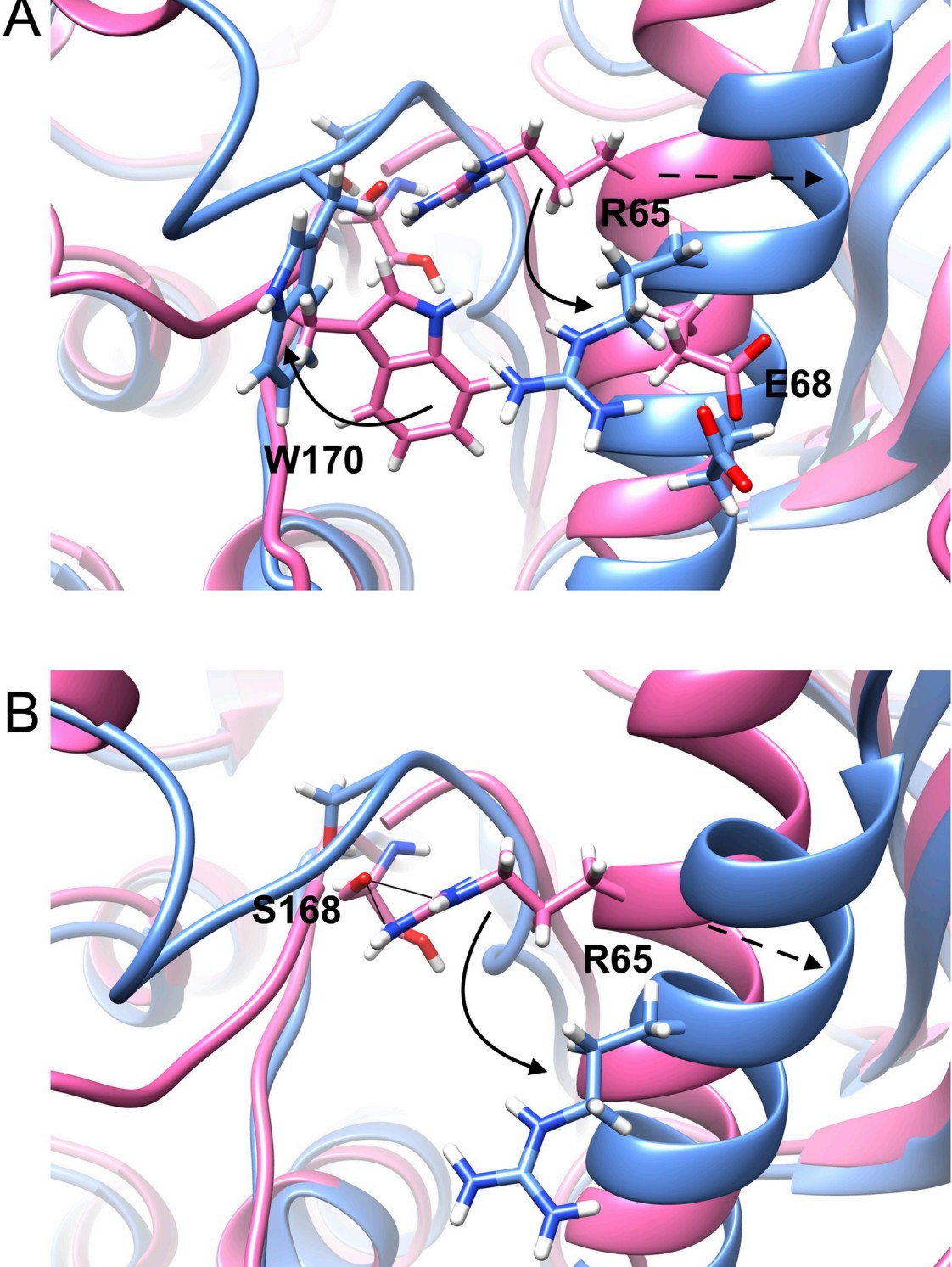

**Fig 11. Structural features differentiating active (magenta) and inactive (blue) states in apo RIPK2.** (A) In active RIPK2, Trp170 orients towards the αC helix measured by the distance between Glu68-CD and Trp170-NE1 and Arg65 forms H-bond interaction with Ser168. (B) Active to inactive transitions breaks the hydrogen bond interaction between Arg65 and Ser168. Flipping of Trp170 destabilizes the activation loop which breaks the H-bond interaction involving Arg65 and Ser168, inducing a shift in αC helix. The solid arrow indicates flipping of Trp170 and relative position of Arg65 in active and inactive RIPK2. The dotted arrow (A, B) indicates shift from inward to outward conformation of αC helix.

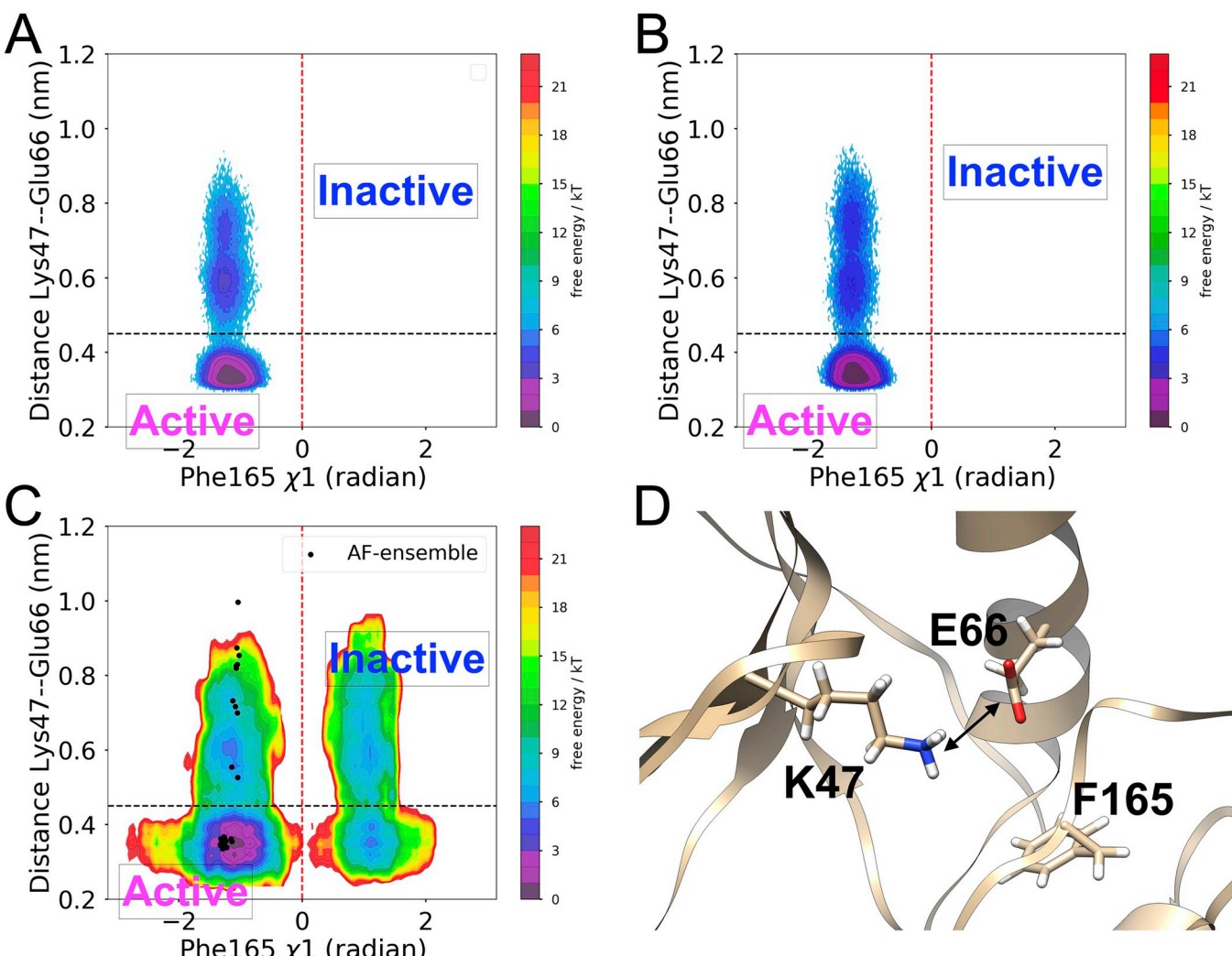

**Fig 12. Active to inactive transition in RIPK2 involves breaking of Lys47-Glu66 salt bridge interaction.** (A) 2D free energy surface along Ph165 $\chi1$ angle and the Lys47—Glu66 distance corresponding unbiased MD simulation of apo RIPK2. (B) Free energy surface projected along Ph165 $\chi1$ angle and the Lys47—Glu66 distance from unbiased MD simulation of RIPK2-XIAP complex highlights stabilization of the active state due to formation of salt-bridge interaction (distance < 0.45 nm) involving Lys47 and Glu66. (C) Reweighted free energy surface from SFA metadynamics simulation sampled transition between active and inactive conformation of RIPK2. Inactive conformation of RIPK2 sampled broken (distance > 0.45 nm) Lys47 and Glu66 salt-bridge interaction along with flipping along $\chi1$ angle of Phe165. Relative orientation of Lys47, Glu66 and Phe165 highlights the broken salt-bridge interaction and flipped Phe165 orientation. The double ended solid arrow highlights the Lys47-Glu66 salt bridge interaction (D). Black dotted line demarcates Lys47—Glu66 distance of 0.45 nm. The red dotted line demarcates Phe165 $\chi1$ angle = 0 radian (C). The black dots (C) correspond to AlphaFold generated conformational ensemble which failed to sample inactive state.

This highlights the importance of performing multiple short MD simulations starting from the AlphaFold-generated conformational ensemble, which not only equilibrates the structures but also samples Boltzmann-weighted transitions across key structural features involved in biologically significant conformational transitions.

Recent works highlighted how we can combine structural ensembles generated by Alpha-Fold with autoencoder framework [50] and MSM [14] to predict Boltzmann distribution and capture rare events. The enhanced sampling strategy we propose, termed SFA-metadynamics, augments the toolkit of approaches aiming to extract the Boltzmann distribution from structural ensembles generated by AlphaFold. This strategy is particularly effective to capture rare

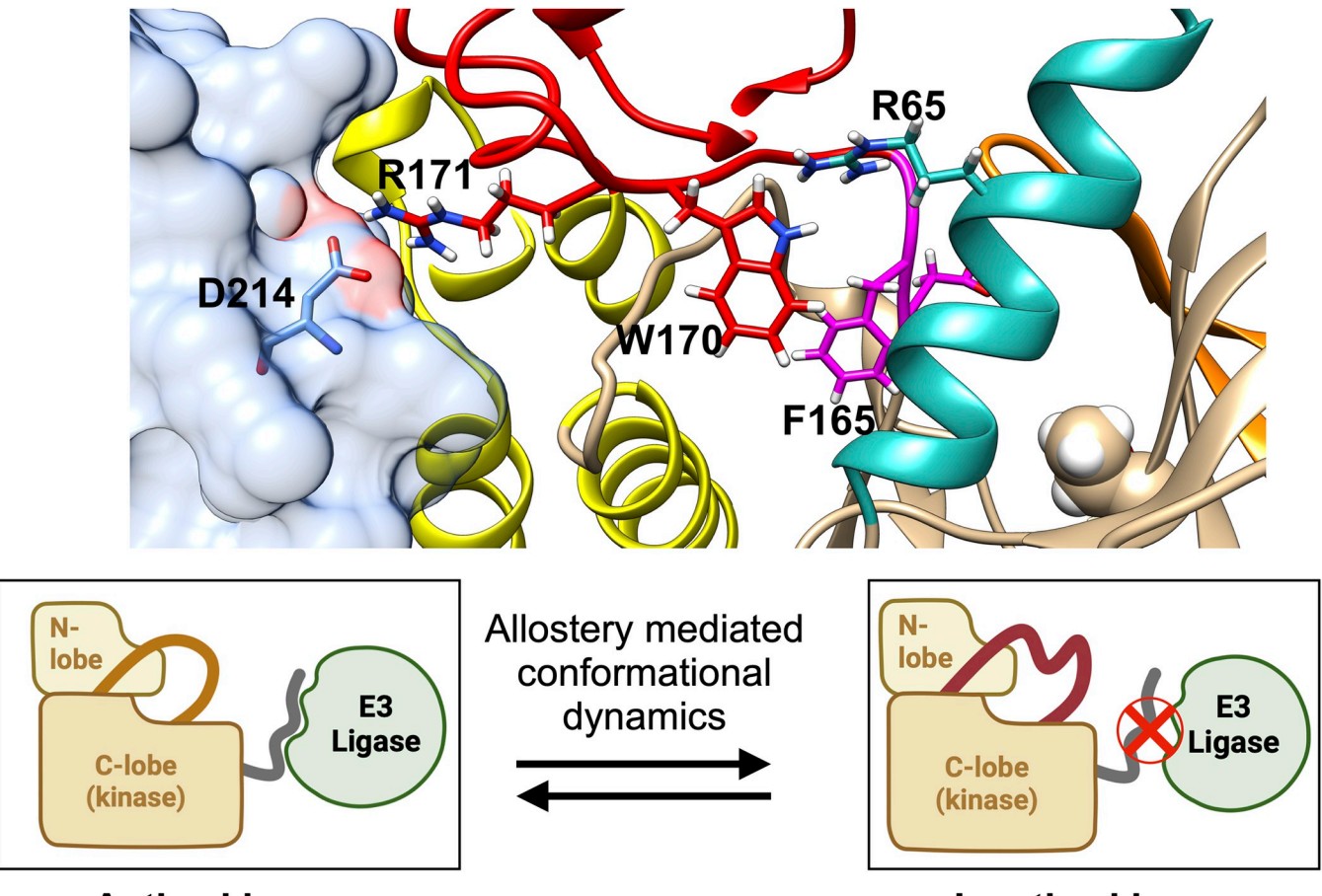

**Fig 13. Mechanism involving conformational dynamics of RIPK2, emphasizing the shift between active and inactive states.** Active state of RIPK2 engages in PPI interactions with the E3 ligase, XIAP, while the inactive state prevents these interactions. Critical residues within the DFG loop (magenta), activation loop (red), and αC helix (sea green) regulate the transition between active and inactive states, thereby influencing the interactions between RIPK2 and XIAP (blue surface).

transitions, such as opening of the 'deep' cryptic pocket in plasmepsin II. SFA in principle is similar to time-structure based independent component analysis (tICA) [51]. However, SFA is based on a straightforward idea: extract uncorrelated output signals that are ordered by slowness. This allows SFA to capture slowly varying features from high-dimensional and noisy temporal data generated by AlphaFold seed molecular dynamics simulations. A key distinction between Time-lagged Independent Component Analysis (TICA) and Slow Feature Analysis (SFA) is that TICA often requires the choice of a lag time and unbiased molecular dynamics simulation data to extract meaningful features. In contrast, SFA does not require the choice of a lag time and can be directly applied to partially converged enhanced sampling data to extract slowly varying features from high-dimensional data. SFA can also act as a reduced space on which one can build MSM. Further, conformations which contribute to the first few slow features can be used as seeds to launch small MD simulations. These MD simulations can be stitched together using MSM to predict thermodynamics and kinetics associated with conformational transitions. Kinetics extracted from MSM built on top of SFA can be directly compared with infrequent metadynamics [52] with slow features as CVs. In essence, SFA emerges as an innovative technique that bridges the gap in the estimation of kinetics linked to rare events, whether one uses MSM or metadynamics.

## Conclusions

In this study, we present a novel integration of slow feature analysis (SFA), derived from Alpha-Fold seeded molecular dynamics (MD) simulations, with metadynamics. This innovative approach is applied to sample cryptic pocket opening, protein-ligand binding/unbinding, and the allosteric dynamics within biomolecules. By synergistically combining SFA with metadynamics, we successfully identified multiple transitions between the closed and open states of the cryptic pocket in plasmepsin II in just a few hundred nanoseconds. This is a significant improvement over traditional methods, which typically require several microseconds of simulation data to develop a comprehensive Markov state model of cryptic pocket opening. Furthermore, the SFA augmented funnel metadynamics not only effectively captured multiple instances of ligand binding and unbinding but also explored essential conformational dynamics of protein that facilitate ligand binding. Conformational transitions in proteins are crucial for ligand binding and significantly influence the kinetics and thermodynamics of the binding process. Previous study [44] has highlighted the use of path CVs to incorporate protein structural dynamics in plasmepsin II, which accelerates the sampling of protein motions necessary for ligand binding. However, path CVs necessitate knowledge 3D structural co-ordinates of starting state, end state, and intermediate states along the transition pathway. Identifying these states is often challenging without prior structural information obtained from extensive experimental data or computational simulations. SFA trained on AlphaFold seeded molecular dynamics captures linear combinations of key structural features accounting for the conformational dynamics of protein without requiring *a priori* information. SFA-augmented funnel metadynamics can serve as a key simulation protocol, effectively capturing the kinetics and thermodynamics of ligand binding by accounting for the conformational plasticity of biomolecules. Additionally, SFA-metadynamics approach discovered the pivotal role played by the activation loop and the DFG moiety in the transition from the active to inactive state in RIPK2. This transition is a critical molecular event that allosterically regulates the interaction between RIPK2 and XIAP, which has implications in the pathogenesis of inflammatory diseases.

This work underscores a novel approach that merges the capabilities of AlphaFold, SFA, and metadynamics. It offers a robust framework to predict the Boltzmann distribution corresponding to conformational shifts necessary to capture rare events such as cryptic pocket opening, protein-ligand binding, and allosteric modulation in biomolecules. The approach of generating diverse conformational ensemble is not limited to AlphaFold but can be extended to other AI based protein structure prediction models such as ESMFold [53], allowing us to extend the applicability of SFA augmented metadynamics beyond AlphaFold. It is important to highlight that in order for methods such as SFA, deep-TICA [54], RAVE [55] to be useful, the training data should transiently sample conformational heterogeneity. Short molecular dynamics simulations launched from AlphaFold generated ensemble encompasses necessary conformational heterogeneity which can be extracted by SFA and used as CVs within enhanced sampling framework such as metadynamics to efficiently sample rare conformational transitions which plays a key role in molecular recognition and conformational dynamics of biomolecules. Finally, AlphaFold-SFA bridges the gap between AI based protein structure prediction model, molecular dynamics simulation and metadynamics to sample rare events of biological relevance, highlighting the importance of understanding conformational dynamics in structure-based drug discovery.

## Supporting information

**S1 Fig. MSM weighted equilibrium population projected along Trp41 $\chi 1$ and $\chi 2$ highlighted sampling of closed and open states of plasmepsin II.** (A)Markov state model

was generated using AlphaFold seeded unbiased molecular dynamics simulations (80 structures * 2 independent clones * 100ns each = total 16 μs) reported by *Meller and coworkers*. MSM was performed on sin and cos transformed $\chi$1 and $\chi$2 angles of Trp41. (B) The convergence of MSM has been highlighted by capturing timescale associated with conformational transition with different choices of clusters (k). Timescales associated with conformational transitions between *open* and *closed* states are in the same order of magnitude highlighting the convergence of MSM.
(PDF)

**S2 Fig. Implied timescales associated with the MSM.** (A) Implied timescale plot associated with Markov state model. We chose lag time of 6ns to generate MSM. (B) *PCCA*$^+$ was used to generate microstate definition associated with Trp41 $\chi$1 and $\chi$2 angles. *PCCA*$^+$ manage to separate closed (*S6*) and open (*S5*) states in plasmepsin II.
(PDF)

**S3 Fig. SFA weights and corresponding features.** (A) SF1 Weights corresponding each feature. (B) SF2 weights corresponding each feature. List of features can be accessed here: https://github.com/sbhakat/AlphaFold-SFA/blob/main/Plm-cryptic-pocket/features.ipynb It highlights how first two slow features (A, B) manage to capture sidechain flipping associated with Trp41 and Tyr77, $\chi$1 and $\chi$2 angles.
(PDF)

**S4 Fig. Projection of SF1 and SF2.** (A) Projection of first two slow features along sin transformed $\chi$1 angle of Trp41 highlights from the training data highlights separation along Trp41, a key residue involved in cryptic pocket opening. (B) Projection of Trp41 $\chi$1 and $\chi$2 angle along SF1. (C) Free energy surface projected on first two slow features (SF1 and SF2) from unbiased MD simulations starting from closed conformation of plasmepsin II. (D) Reweighted free energy surface projected along first two slow features from metadynamics simulations with slow features as CVs highlights accelerated sampling compared to unbiased MD simulations.
(PDF)

**S5 Fig. Projection of first two slow features for plasmepsin-II.** (A) Time trace SF1 in the training data. (B) Time traced of SF1 in SFA-metadynamics. (C) Time trace of SF2 in the training data. (D) Time trace of SF2 in SFA-metadynamics. Metadynamics simulations using slow features as CVs manage to capture multiple recrossing within a few hundreds of nanoseconds (C, D).
(PDF)

**S6 Fig. Converge of SFA-metadynamics simulation in plasmepsin-II.** (A-E) Reweighted free energy surfaces from well-tempered metadynamics projected along Trp41 $\chi$1 and $\chi$2 angles at different time intervals (A-E) highlighted convergence of our metadynamics simulations. (F) Time trace of the reweighting factor *rc(t)* also highlighted the convergence of metadynamics. It is important to highlight that the reweighting factor reached an asymptotic plateau at the end of the metadynamics simulations which highlights the convergence of the simulation.
(PDF)

**S7 Fig. Volume of cryptic pocket in plasmepsin-II.** (A) Space fill representation of binding pocket in closed plasmepsin-II (PDB: 1LF4). (B) Space fill representation of deep cryptic pocket opening (open, PDB: 2BJU) increases the volume of the binding pocket compared to closed conformation of plasmepsin-II. CASTp server (http://sts.bioe.uic.edu/castp/index.html?3trg) predicts the volume of open and closed conformation are 1185.592 and 391.228 Å$^3$

respectively.
(PDF)

**S8 Fig. Reweighted free energy surface captures flap opening and flipping of Tyr77 in plasmepsin-II.** (A) Free energy surface projected along flap opening and Tyr77 $\chi1$ angle for unbiased MD simulations starting from closed state. (B) Reweighted free energy surface projected along flap opening and Tyr77 $\chi1$ angle from SFA-metadynamics starting from closed state. (C) Free energy surface projected along flap opening and Tyr77 $\chi1$ angle for unbiased MD simulations starting from open state. (D) Reweighted free energy surface projected along flap opening and Tyr77 $\chi1$ angle from SFA-metadynamics starting from open state. SFA-metadynamics captures flipping of Tyr77 $\chi1$ angle and flap opening within few hundreds of nanoseconds when compared to 16 μs of unbiased molecular dynamics simulation of started from closed and open states respectively. AlphaFold generated conformations are shown as black dots.
(PDF)

**S9 Fig. SFA-metadynamics samples alternative cryptic pocket open states in plasmepsin-II.** (A) Conformation of cryptic pocket *open* state with Trp41 $\chi2$ angle of +1 radian (orange, PDB: 2BJU). (B) Conformation of cryptic pocket *open* state with Trp41 $\chi2$ angle of -1 radian (blue, PDB: 4Z22). (C) Reweighted free energy surface from SFA-metadynamics along Trp41 $\chi1$ and $\chi2$ angles highlighted alternate states (A, B) associated with $\chi2$ flipping. AlphaFold ensembles are highlighted in black dots. It is important to note that AlphaFold failed to sample an alternate Trp41 $\chi2$ angle of -1 radian. The black dotted arrows highlight the flipping of Trp41 and the flap opening.
(PDF)

**S10 Fig. SFA-augmented funnel metadynamics captures protein dynamics of plasmepsin-II.** (A) Snapshot from traditional funnel metadynamics (blue) highlights the formation of H-bond interaction between Tyr77 and Asp34 which blocks deep cryptic pocket in plasmepsin-II when compared to the starting conformation (magenta, PDB: 7QYH). (B) Time trace of H-bond interaction between Tyr77-Asp34 in traditional funnel metadynamics. (C) Time trace of H-bond interaction between Tyr77-Asp34 in SFA-augmented funnel metadynamics.
(PDF)

**S11 Fig. SFA weights and corresponding features in RIPK2.** (A) SF1 weights and corresponding sin/cos transformed dihedral angles. (B) SF2 weights and corresponding sin/cos transformed dihedral angles (https://github.com/sbhakat/AlphaFold-SFA/blob/main/RIPK2/feature-lists.ipynb) associated with amino acid residues in RIPK2.
(PDF)

**S12 Fig. Sampling of first two slow features in the training data and metadynamics simulations starting with apo RIPK2.** (A) Time trace SF1 in the training data. (B) Time traced of SF1 in SFA-metadynamics. (C) Time trace of SF2 in the training data. (D) Time trace of SF2 in SFA-metadynamics. Metadynamics accelerated the sampling along first two slow features which enabled sampling of allosteric dynamics in RIPK2.
(PDF)

**S13 Fig. Convergence of SFA metadynamics starting with apo RIPK2.** (A-D) Reweighted free energy surfaces from well-tempered SFA-metadynamics projected along Phe165 $\chi1$ and $\chi2$ angles at different time intervals highlighted convergence of the simulation.
(PDF)

**S14 Fig. SFA metadynamics samples active and inactive conformation of RIPK2.** Distribution of Phe165 dihedral angles in RIPK2 crystal structures available in RCSB PDB projected on reweighted free energy surface from SFA-metadynamics. Inactive RIPK2 corresponds to PDB: 5J7B, 5AR4, 6SZJ and 6FU5.
(PDF)

**S15 Fig. SFA-metadynamics samples multiple transitions associated with H-bond involving Arg65 and Ser168.** (A) Time trace of Arg65—Ser168 distance during SFA-metadynamics. (B) Time trace of Arg65—Ser168 distance in unbiased MD simulation of apo RIPK2. SFA-metadynamics managed to sample multiple transitions between active and inactive states of RIPK2 (demarked by the dashed line at 0.82 nm) compared to unbiased MD simulation which remained in the active state.
(PDF)

**S16 Fig. SFA metadynamics captures inward to outward transition in apo RIPK2.** (A) Time trace of RMSD of αC helix in unbiased MD. (B) Time trace of RMSD of αC helix in SFA-metadynamics simulation. In unbiased MD simulations αC helix remained in an *inward* conformation due to H-bond interaction involving Arg65—Ser168. SFA-metadynamics simulations managed to capture conformational dynamics associated with the activation loop of RIPK2 which sampled flipping of Trp170. Flipping of Trp170 destabilizes the activation loop which breaks Arg65—Ser168 interaction and resulted in an *outward* conformation of αC helix.
(PDF)

**S17 Fig. RIPK2-XIAP complex stabilizes the active state of RIPK2.** (A) Free energy surface projected along $\chi1$ and $\chi2$ angles of Trp170 from unbiased MD simulation of RIPK2+XIAP complex. (B) Free energy surface projected along $\chi1$ and $\chi2$ angles of Phe165 from unbiased MD simulation of RIPK2+XIAP complex. (C) Time trace of Arg65—Ser168 distance in unbiased MD simulation of RIPK2+XIAP complex. (D) Time trace of Glu68—Trp170 distance in unbiased MD simulation of RIPK2+XIAP complex. (E) Time trace of Arg171—XIAP-Asp214 distance in unbiased MD simulation of RIPK2+XIAP complex.
(PDF)

**S18 Fig. Structural insights of how Arg171 present in the activation loop of RIPK2 interacts with the Asp214 of XIAP (green).** It highlights orientation of Phe165, Trp170, Arg65 and Ser168, key residues involved in conformational dynamics of RIPK2.
(PDF)

**S19 Fig. Conformational sampling in AlphaFold generated ensemble of apo RIPK2.** Sampling of Phe165 (A), Trp170 (B) and Arg65 (C) in AlphaFold generated conformational ensemble of apo RIPK2.
(PDF)

**S20 Fig. Active to inactive transition has been sampled in homologous serine-threonine kinase, BRAF.** (A) Crystal structures of holo-BRAF highlights the coupled motion involving DFG-Phe flipping and the 'outward' conformation of αC helix. Blue indicates PDB: 4EHG and the magenta indicates PDB: 2FB8. (B) Outward conformation of αC helix breaks the salt bridge interaction between Lys483 and Glu501. Breaking of the salt bridge interaction and flipping of Phe595 is a hall mark of Src-like inactive conformation.
(PDF)

**S21 Fig. Interaction between Lys209 of RIPK2 and Glu211 of XIAP is critical for the formation of protein-protein complex.** (A) Time-trace of distance between Lys209 of RIPK2 and Glu211 of XIAP during total 3 μs of unbiased MD simulation of XIAP-RIPK2 complex. The dashed line at 0.35 nm indicates formation of H-bond interaction involving Lys209—Glu211. (B) Orientation of Lys209 and Ile208 of RIPK2 and Glu211 of XIAP is highlighted for visual inspection.
(PDF)

**S22 Fig. Protein-protein interface of RIPK2-XIAP complex highlighting key residues involved in H-bond interactions.** The activation loop of the RIPK2 is highlighted in red and the XIAP is highlighted in 'seagreen blue'.
(PDF)

**S23 Fig. Time trace of key interactions involving the interface of RIPK2-XIAP complex during unbiased MD simulations.** (A) Time trace of RIPK2-Arg280—XIAP-Asn209 distance. (B) Time trace of RIPK2-Lys285—XIAP-Asp196 distance. (C) Time trace of RIPK2-Glu279—XIAP-Lys208 distance. Temporal evolution of hydrogen bond distances throughout molecular dynamics simulations, encompassing 15 independent runs each lasting 200 nanoseconds, across crucial residues that play a role in stabilizing the RIPK2-XIAP complex. The dotted line at 0.35 nm indicates complete formation of H-bond.
(PDF)

**S1 Text. Choice of sequences for structure prediction.**
(PDF)

## Acknowledgments

Molecular dynamics and metadynamics simulation corresponding plasmepsin and RIPK2 were performed on computer resources provided by the Swedish National Infrastructure for Computing (SNIC) at LUNARC (Lund University) and HPC2N (Umeå University). Funnel metadynamics calculations were partly performed in Riga Technical University (RTU) High Performance Computing (HPC) Centre facilities.

## Author Contributions

**Conceptualization:** Soumendranath Bhakat.

**Data curation:** Shray Vats, Raitis Bobrovs, Pär Söderhjelm, Soumendranath Bhakat.

**Formal analysis:** Shray Vats, Raitis Bobrovs.

**Methodology:** Shray Vats.

**Resources:** Raitis Bobrovs, Pär Söderhjelm.

**Software:** Shray Vats.

**Supervision:** Soumendranath Bhakat.

**Validation:** Shray Vats.

**Writing – original draft:** Soumendranath Bhakat.

**Writing – review & editing:** Shray Vats, Soumendranath Bhakat.

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
