## [Decision Letter · Decision Letter 0]

12 Jun 2024

PONE-D-24-21616AlphaFold-SFA: accelerated sampling of cryptic pocket opening, protein-ligand binding and allostery by AlphaFold, slow feature analysis and metadynamicsPLOS ONE

Dear Dr. Bhakat,

Thank you for submitting your manuscript to PLOS ONE. After careful consideration, we feel that it has merit but does not fully meet PLOS ONE’s publication criteria as it currently stands. Therefore, we invite you to submit a revised version of the manuscript that addresses the points raised during the review process.

We look forward to receiving your revised manuscript.

Kind regards,

Xiakun Chu, Ph.D.

Academic Editor

PLOS ONE

Journal Requirements:

2. Please note that PLOS ONE has specific guidelines on code sharing for submissions in which author-generated code underpins the findings in the manuscript. In these cases, all author-generated code must be made available without restrictions upon publication of the work. 

Please review our guidelines at https://journals.plos.org/plosone/s/materials-and-software-sharing#loc-sharing-code and ensure that your code is shared in a way that follows best practice and facilitates reproducibility and reuse.

"Authors declare no conflict of interests. Soumendranath Bhakat and Shray Vats are cofounders of AlloTec Bio, Inc."

We note that one or more of the authors are employed by a commercial company: AlloTec Bio, Inc.

2) Please also provide an updated Competing Interests Statement declaring this commercial affiliation along with any other relevant declarations relating to employment, consultancy, patents, products in development, or marketed products, etc.  

Within your Competing Interests Statement, please confirm that this commercial affiliation does not alter your adherence to all PLOS ONE policies on sharing data and materials by including the following statement: ""This does not alter our adherence to  PLOS ONE policies on sharing data and materials.” (as detailed online in our guide for authors http://journals.plos.org/plosone/s/competing-interests) . If this adherence statement is not accurate and  there are restrictions on sharing of data and/or materials, please state these. Please note that we cannot proceed with consideration of your article until this information has been declared.

4. Please note that your Data Availability Statement is currently missing the repository name. If your manuscript is accepted for publication, you will be asked to provide these details on a very short timeline. We therefore suggest that you provide this information now, though we will not hold up the peer review process if you are unable.

Reviewers' comments:

Reviewer's Responses to Questions

**Comments to the Author**

1. Is the manuscript technically sound, and do the data support the conclusions?

Reviewer #1: Yes

Reviewer #2: Partly

Reviewer #3: Partly

2. Has the statistical analysis been performed appropriately and rigorously? 

Reviewer #1: Yes

Reviewer #2: Yes

Reviewer #3: Yes

3. Have the authors made all data underlying the findings in their manuscript fully available?

Reviewer #1: Yes

Reviewer #2: Yes

Reviewer #3: No

4. Is the manuscript presented in an intelligible fashion and written in standard English?

Reviewer #1: Yes

Reviewer #2: Yes

Reviewer #3: Yes

5. Review Comments to the Author

Reviewer #1: Comment of the referee: “In this study, the authors provide insight into how SFA acts as a dimensionality reduction tool that bridges the gap between AlphaFold, molecular dynamics simulation, and metadynamics in the context of capturing rare events in biomolecules, extending the scope of structure-based drug discovery in the era of AlphaFold. The work is of good quality, and I recommend publishing it after addressing the minor changes below.”

1. Alphafold is a good structure prediction tool, but what is the advantage of using the conformations it generates as the starting conformations to generate CV compared to using a simple Steer MD and path cv? If it is just for taking rare conformations, some high-energy unreasonable conformations themselves may also have a bad effect on the prediction of CV. This part may require more explanation.

2. This method uses microsecond-level calculations when generating CV. If the corresponding computing resources are directly used to calculate metadynamics, can better convergence be achieved? What are the specific applications and advantages of this method?

3. It is easy to see the acceleration of this method in sampling, but the convergence effect does not seem to be reflected. The results in the article are all variables with specific physical meanings, such as in Figure 9 (B, D). Is this the result of reweighting after the selected CV converges?

Reviewer #2: In this paper, the authors developed slow feature analysis (SFA)18, an unsupervised learning algorithm designed to capture slowly varying features from high-dimensional temporal data generated by MD simulations. SFA is utilized as a dimensionality reduction algorithm that bridges the gap between AlphaFold and metadynamics, effectively capturing the Boltzmann distribution associated with a wide range of biological phenomena. These phenomena include cryptic pocket opening, protein-ligand binding, and the identification of allosteric hotspots involved in kinase-mediated protein-protein interactions. The results presented are quite interesting and biologically significant. However, I have several comments that need to be addressed:

1.The paper should introduce the biological background of cryptic pockets to enhance understanding for general readers.

2. Clarification is needed on which sequences were selected to generate the structural ensemble using the ColabFold implementation of AlphaFold. The rationale for selecting these sequences and their biological relevance should be provided.

3. Several figures lack clarity, hindering the assessment of certain conclusions:

In Figure 10, Arg65 forms an H-bond interaction with Ser168, but Ser168 is not labeled.

In Figure 8, the key residues are not clearly labeled.

In Figure 11 (D), the location of the salt-bridge interaction is not clearly indicated.

Reviewer #3: In this study, Vats et al. combined several methods, such as AF structure prediction, high-dimensional reduction, and enhanced sampling methods to study the cryptic pocket opening and protein-ligand binding of receptor-interacting protein kinase2. They performed exhaustive computation and analysis on the kinase system. It is well writen and good organized. However, many big concerns should be addressed.

1. While utilizing AI or AlphaFold to predict protein structures is a research trending area, but it is not closely related to the main thrust of this paper and does not play a significant role in it. AlphaFold (AF) performs structural predictions based on experimental data and known information, but the structures within the ensemble it produces may not have physical meaning. What is the significance of using these structures as initial structures for further MD simulations? Alternatively, the author should compare the differences and advantages, if any, between computational simulations using a single AF predicted structure versus an ensemble of structures as the initial structure. Otherwise, it is completely unnecessary.

2. This paper applies the method of Slow Feature Analysis (SFA) to analyze the results of unbiased MD simulations and uses the so-called "slow features" obtained as reaction coordinates for metadynamics. Let's set aside whether this method can truly capture slow features or not. The reduced-dimensional feature values obtained from SFA do not inherently possess physical meanings. The author needs to provide further details on how to add a Gaussian potential to this Collective Variable (CV) in metadynamics and conduct subsequent sampling.

3. In the Funnel metadynamics based on SFA features, the author did not set a reaction coordinate for the distance between the ligand and the protein. How can we ensure that rapid sampling can be achieved and the ligand-protein binding process can be derived?

4. In the Funnel Metadynamics (FM) based on SFA features, the SFA features are obtained by learning the system without ligands. How can we ensure that the changes in the protein and the reaction coordinates after introducing ligands are not affected by the ligands?

5. What is the connection between the features learned by the author through SFA and the reaction coordinates used to construct the free energy surface in Figure 4/5? Why not create a free energy surface projected onto the SFA features?

6. PLOS authors have the option to publish the peer review history of their article (what does this mean?). If published, this will include your full peer review and any attached files.

Reviewer #1: No

Reviewer #2: No

Reviewer #3: No

---

## [Author Response · Author response to Decision Letter 0]

23 Jun 2024

Response to the Reviewers

The authors appreciate fast review process and constructive comments by the PlosOne academic editor and the reviewers. We addressed each reviewer comments and added modifications/clarifications accordingly in the main text and the supplementary information (highlighted in BLUE). We thank the academic editor and the reviewers for their time and effort to enhance our manuscript.

Reviewer #1: 

1. Alphafold is a good structure prediction tool, but what is the advantage of using the conformations it generates as the starting conformations to generate CV compared to using a simple Steer MD and path cv? If it is just for taking rare conformations, some high-energy unreasonable conformations themselves may also have a bad effect on the prediction of CV. This part may require more explanation.

Answer: This is a great point. Bobrovs and coworkers previously used path CV in conjunction with traditional distance CVs in funnel metadynamics to account for protein dynamics (reference 41, paper: https://pubs.acs.org/doi/10.1021/acs.jcim.2c00422 ). However, setting up a path-based CV is challenging as it needs 3D-co-ordinates of starting state, end state and intermediate states. Bobrovs and coworkers used existing crystal structures of plasmepsin II to set up a path CV which better explores flap dynamics necessary for ligand binding. However, SFA is a compressed representation of multiple CVs which does not need prior information of conformational states similar to path CVs. SFA trained on molecular dynamics simulations data captures underlying key features governing flap dynamics and cryptic pocket opening in plasmepsin II as highlighted in Figure 4 and 5. This makes SFA a robust alternative of path CV and allows more flexibility to incorporate within metadynamics protocol. We have added the following section in the manuscript: 

“Previous study41 has highlighted the use of path CVs to incorporate protein structural dynamics, which accelerates the sampling of protein motions necessary for ligand binding. However, path CVs necessitate knowledge 3D structural co-ordinates of starting state, end state, and intermediate states along the transition pathway. Identifying these states is often challenging without prior structural information obtained from extensive experimental data or computational simulations. SFA trained on AlphaFold seeded molecular dynamics captures linear combinations of key structural features accounting for the conformational dynamics of protein without requiring a priori information.”

We would also like to highlight that steer MD is not a natural descriptor of complex biological motion highlighted in the manuscript which is a function of multiple degrees of freedom across different features. Steer MD would allow us to see ligand binding however it would fail to capture flap opening + Trp41, two key conformational events necessary to effectively sample multiple transitions of ligand unbinding and binding. 

Authors raised a great question: Is it possible that incorporating rare high energy conformations has bad effect in prediction of CV? This is a great point, and the answer is YES. If we trained SFA directly on AlphaFold output, we would have incorporated random noise in our CV prediction. Molecular dynamics simulations initiated from AlphaFold structures not only ensure that the conformations are well-equilibrated but also guarantee that the training data for SFA represents an unbiased Boltzmann distribution.

We draw attention to Figures 9 (A, B), 10 and 12C, where the black dots represent AlphaFold-generated structures. It is evident that AlphaFold alone failed to sample active-to-inactive transitions. Had we trained SFA exclusively on AlphaFold output, we would have missed the key features necessary for capturing active-to-inactive transitions in apo RIPK2. The subsampling of AlphaFold structures introduces conformational diversity, which, when combined with molecular dynamics simulations, enables the sampling of essential structural motions. SFA then identifies these key structural motions as linear combinations.

The capacity to detect multidimensional structural features, often absent in AlphaFold output, allows SFA to serve as an effective CV within the metadynamics framework, thereby accelerating conformational sampling. 

“SFA trained on molecular dynamics simulations seeded from AlphaFold structures captures Boltzmann-weighted structural features necessary to sample conformational transitions associated with cryptic pocket opening in plasmepsin II and allosteric dynamics in RIPK2. In the case of cryptic pocket opening, initiating multiple short unbiased molecular dynamics simulations from the diverse conformational ensemble generated by AlphaFold confers an advantage in conformational sampling, as this ensemble captures multiple degrees of freedom along key structural features (Figures 4 and 5).

Conversely, for RIPK2, AlphaFold alone failed to sample conformational transitions across key structural features involved in active-to-inactive transitions (Figures 9, 10, and 12). SFA trained solely on the AlphaFold-generated ensemble would have been unable to capture the critical transitions required for active-to-inactive transitions. However, short molecular dynamics simulations initiated from the AlphaFold-generated ensemble of RIPK2 which sampled structural heterogeneity of the activation loop (missing in the crystal structures of apo and holo RIPK2), captures Boltzmann-weighted structural features governing active-to-inactive transitions.

This highlights the importance of performing multiple short MD simulations starting from the AlphaFold-generated conformational ensemble, which not only equilibrates the structures but also samples Boltzmann-weighted transitions across key structural features involved in biologically significant conformational transitions.” 

2. This method uses microsecond-level calculations when generating CV. If the corresponding computing resources are directly used to calculate metadynamics, can better convergence be achieved? What are the specific applications and advantages of this method?

Answer: It is well known that longer you perform metadynamics you will reach a asymptotic convergence limit (https://pubs.aip.org/aip/jcp/article-abstract/143/23/234112/193998/A-perturbative-solution-to-metadynamics-ordinary?redirectedFrom=fulltext ). To develop a fully connected Markov State Model (MSM) for estimating the free energy surface (FES) along key features, we would have required approximately 16 microseconds of unbiased molecular dynamics simulation data, as illustrated in Figure S1 of the Supplementary Information. In contrast, SFA-metadynamics enables FES prediction within a few hundred nanoseconds, underscoring its significant advantage over methods like MSM. Moreover, MSM itself cannot be directly incorporated into enhanced sampling workflows.

SFA offers the dual benefit of allowing the construction of an MSM based on a compressed representation of multidimensional features while also serving as a collective variable (CV) within enhanced sampling frameworks such as metadynamics. This versatility extends its applicability as a unifying approach, bridging MSM and enhanced sampling methodologies. We have added the following in the discussion to highlight that

“SFA can also act as a reduced space on which one can build MSM. Further, conformations which contribute to the first few slow features can be used as seeds to launch small MD simulations. These MD simulations can be stitched together using MSM to predict thermodynamics and kinetics associated with conformational transitions. Kinetics extracted from MSM built on top of SFA can be directly compared with infrequent metadynamics51 with slow features as CVs. In essence, SFA emerges as an innovative technique that bridges the gap in the estimation of kinetics linked to rare events, whether one uses MSM or metadynamics.”

In our forthcoming paper, we will demonstrate how SFA also serves as an effective tool for identifying key structural features to initiate adaptive sampling simulations, thereby uncovering conformational allostery upon ligand binding.

3. It is easy to see the acceleration of this method in sampling, but the convergence effect does not seem to be reflected. The results in the article are all variables with specific physical meanings, such as in Figure 9 (B, D). Is this the result of reweighting after the selected CV converges?

Answer: The FES was projected along different features after reweighting. This is highlighted in the main text: “Unbiased free energy surfaces along different features were extracted using the reweighting protocol developed by Tiwary and Parrinello37.”

The convergence along CVs for plasmepsin II and RIPK2 has been described in detail in the Supplementary Information. We have highlighted this in the main text as follows: “Comparison of reweighted free energy surfaces at different time intervals from metadynamics simulations highlighted the convergence (Figure S6 in Supporting Information) for the choice of force field and water model.”

Accelerated sampling along first two slow features (Figure S5 in Supporting Information) , reweighted free energy surface along Trp41 *χ*1 and *χ*2 along different time intervals (Figure S6 in Supporting Information) and projection of the metadynamics reweighting factor (see more here: https://pubs.acs.org/doi/10.1021/jp504920s ) as a function of simulation time clearly highlights the relative convergence of the metadynamics simulations within the simulation time. Similar projections have been highlighted for RIPK2 systems in Figure S12 and S13 of the Supplementary Information. 

Reviewer 2:

1.The paper should introduce the biological background of cryptic pockets to enhance understanding for general readers.

Answer: We have introduced the background of cryptic pocket and allostery in the main text. Highlighted below:

“Cryptic pockets have emerged as a frontier in modern drug discovery, offering new opportunities to target proteins previously considered "undruggable." These hidden binding sites are not apparent in the protein's apo state but become accessible upon interaction with small molecules or during rare conformational transitions. Sampling cryptic pocket opening in apo proteins is of significant interest in drug discovery, as targeting these pockets has the potential to yield highly selective and potent inhibitors with unique structural features distinct from catalytic sites or known binding interfaces. The strategy of targeting cryptic pockets has proven attractive in developing selective inhibitors against challenging targets, such as plasmepsin-II, a key enzyme in antimalarial drug discovery. Similar to cryptic pockets, allostery in protein kinases is a rare conformational event which represents a complex signaling pathway via which changes in the active site of kinase induces conformational changes at a distal site. Understanding allostery in kinases is key to uncovering the mechanisms by which these enzymes interact with protein partners and regulate downstream signaling across several disease pathways. Elucidating allosteric fingerprints is key to gain insights into kinase function beyond catalytic activity, potentially revealing new strategies for therapeutic intervention in various pathological conditions.”

2. Clarification is needed on which sequences were selected to generate the structural ensemble using the ColabFold implementation of AlphaFold. The rationale for selecting these sequences and their biological relevance should be provided.

Answer: We have added a new section titled “Choice of sequences for structure prediction:” in the Supplementary Information. We have added the following in the main text:

“The specific sequences used to generate the conformational ensemble are detailed in the 'Choice of sequences for structure prediction' section of the Supporting Information.”

3. Several figures lack clarity, hindering the assessment of certain conclusions:

In Figure 10, Arg65 forms an H-bond interaction with Ser168, but Ser168 is not labeled.

Answer: We have introduced a new Figure 11 (previous numbering Figure 10) highlighting the Arg65 and Ser168 H-bond and relative orientation of the key residues. 

In Figure 8, the key residues are not clearly labeled.

We have added the label for Phe165 in the updated Figure 9 (previously Figure 8). 

In Figure 11 (D), the location of the salt-bridge interaction is not clearly indicated.

Highlighted as a double ended arrow in the modified Figure 12 (D) (previously Figure 11)

Reviewer 3:

1. While utilizing AI or AlphaFold to predict protein structures is a research trending area, but it is not closely related to the main thrust of this paper and does not play a significant role in it. AlphaFold (AF) performs structural predictions based on experimental data and known information, but the structures within the ensemble it produces may not have physical meaning. What is the significance of using these structures as initial structures for further MD simulations? Alternatively, the author should compare the differences and advantages, if any, between computational simulations using a single AF predicted structure versus an ensemble of structures as the initial structure. Otherwise, it is completely unnecessary.

Answer: To test reviewer’s hypothesis, we generated conformation from the AlphaFold3 web server (https://alphafoldserver.com) and performed combined 3 microsecond of unbiased molecular dynamics simulation and projected the distribution along Trp41 *χ*1 and *χ*2 angles. It is clear that starting unbiased molecular dynamics simulations from closed conformation fails to capture open < -- > close transition as highlighted in Figure 4 (A, C) of the manuscript.

We will like to highlight the reviewer two of the key papers in this field which describe in the detail the sampling benefit using the stochastic subsampling of AlphaFold’s input MSA: Paper 1: https://pubs.acs.org/doi/epdf/10.1021/acs.jctc.2c01189 and 

Paper 2: https://pubs.acs.org/doi/abs/10.1021/acs.jctc.3c00290

 See the attached document for the Figure.

Figure. A: Conformations generated by AlphaFold 3 webserver (https://alphafoldserver.com ), B: Trp41 adapts closed conformation in the conformations generated by AlphaFold 3 webserver. C: Projection of unbiased free energy surface along Trp41 *χ*1 and *χ*2 angles highlight lack of sampling of cryptic pocket open states (PDB: 2BJU, 4Z22). 

We have also added the following section in the revised manuscript 

“SFA trained on molecular dynamics simulations seeded from AlphaFold structures captures Boltzmann-weighted structural features necessary to sample conformational transitions associated with cryptic pocket opening in plasmepsin II and allosteric dynamics in RIPK2. In the case of cryptic pocket opening, initiating multiple short unbiased molecular dynamics simulations from the diverse conformational ensemble generated by AlphaFold confers an advantage in conformational sampling, as this ensemble captures multiple degrees of freedom along key structural features (Figures 4 and 5).

Conversely, for RIPK2, AlphaFold alone failed to sample conformational transitions across key structural features involved in active-to-inactive transitions (Figures 9, 10, and 12). SFA trained solely on the AlphaFold-generated ensemble would have been unable to capture the critical transitions required for active-to-inactive transitions. However, short molecular dynamics simulations initiated from the AlphaFold-generated ensemble of RIPK2 which sampled structural heterogeneity of the activation loop (missing in the crystal structures of apo and holo RIPK2), captures Boltzmann-weighted structural features governing active-to-inactive transitions.

This highlights the importance of performing multiple short MD simulations starting from the AlphaFold-generated conformational ensemble, which not only equilibrates the structures but also samples Boltzmann-weighted transitions across key structural features involved in biologically significant conformational transitions.”

2. This paper applies the method of Slow Feature Analysis (SFA) to analyze the resu

---

## [Decision Letter · Decision Letter 1]

2 Jul 2024

AlphaFold-SFA: accelerated sampling of cryptic pocket opening, protein-ligand binding and allostery by AlphaFold, slow feature analysis and metadynamics

PONE-D-24-21616R1

Dear Dr. Bhakat,

We’re pleased to inform you that your manuscript has been judged scientifically suitable for publication and will be formally accepted for publication once it meets all outstanding technical requirements.

Kind regards,

Xiakun Chu, Ph.D.

Academic Editor

PLOS ONE

Additional Editor Comments (optional):

I thank authors for their great efforts in this manuscript. Reviewer 2 seemed to miss the SI, I have checked the SI, which is all right for publication.

Reviewers' comments:

Reviewer's Responses to Questions

**Comments to the Author**

1. If the authors have adequately addressed your comments raised in a previous round of review and you feel that this manuscript is now acceptable for publication, you may indicate that here to bypass the “Comments to the Author” section, enter your conflict of interest statement in the “Confidential to Editor” section, and submit your "Accept" recommendation.

Reviewer #1: All comments have been addressed

Reviewer #2: (No Response)

Reviewer #3: All comments have been addressed

2. Is the manuscript technically sound, and do the data support the conclusions?

Reviewer #1: Yes

Reviewer #2: Yes

Reviewer #3: Yes

3. Has the statistical analysis been performed appropriately and rigorously? 

Reviewer #1: Yes

Reviewer #2: Yes

Reviewer #3: Yes

4. Have the authors made all data underlying the findings in their manuscript fully available?

Reviewer #1: Yes

Reviewer #2: Yes

Reviewer #3: Yes

5. Is the manuscript presented in an intelligible fashion and written in standard English?

Reviewer #1: Yes

Reviewer #2: Yes

Reviewer #3: Yes

6. Review Comments to the Author

Reviewer #1: The author clearly explains the advantages of using alphafold compared to traditional steer MD and path cv, and at the same time completes the data on convergence and answers the questions raised. The article has a large workload and provides some new methods for solving some unsolved problems such as ligand binding. I think this article meets the requirements for publication in this journal.

Reviewer #2: Comments 1 and 3 have been addressed, for Comments 2, I did not find a new section titled “Choice of sequences for structure prediction:” in the Supplementary Information. We just find a Supplementary Information which only include figures.

Reviewer #3: Now it is publishable.

7. PLOS authors have the option to publish the peer review history of their article (what does this mean?). If published, this will include your full peer review and any attached files.

Reviewer #1: No

Reviewer #2: No

Reviewer #3: No

---

## [Editor Report · Acceptance letter]

10 Jul 2024

PONE-D-24-21616R1 

PLOS ONE

Dear Dr. Bhakat, 

I'm pleased to inform you that your manuscript has been deemed suitable for publication in PLOS ONE. Congratulations! Your manuscript is now being handed over to our production team.

Kind regards, 

on behalf of

Dr. Xiakun Chu 

Academic Editor

PLOS ONE